# Achieving Global Flatness in Decentralized Learning with Heterogeneous Data

**Sakshi Choudhary**                                                                 *choudh23@purdue.edu*
*Department of Electrical and Computer Engineering*
*Purdue University*

**Sai Aparna Aketi**                                                                   *aketiaparna@gmail.com*
*Central Applied Science*
*Meta*

**Kaushik Roy**                                                                        *kaushik@purdue.edu*
*Department of Electrical and Computer Engineering*
*Purdue University*

**Reviewed on OpenReview:** *https://openreview.net/forum?id=8G32T4RLbX*

## Abstract

Decentralized training enables peer-to-peer on-device learning without relying on a central server, but suffers from degraded generalization performance under heterogeneous data distributions due to local overfitting. One strategy to mitigate this is to seek flatter loss landscapes during local optimization at each client. However, with extreme data heterogeneity, local objectives may diverge from the global one, yielding local flatness rather than true global flatness. To mitigate this challenge, we introduce GFlat, a novel decentralized algorithm that enables each client to estimate and incorporate an approximation of the global update direction while seeking a flatter loss landscape locally. This lightweight strategy allows each client to directly contribute to global flatness without requiring additional communication or centralized coordination. We theoretically analyze the convergence properties of GFlat and validate its performance through extensive experiments across a range of datasets, model architectures, and communication topologies. GFlat consistently improves generalization in non-IID data settings and achieves up to 6.75% higher test accuracy compared to state-of-the-art decentralized methods. [1]

## 1 Introduction

Modern deep neural networks are fueled by massive amounts of training data generated at edge devices such as smartphones, Internet-of-Things (IoT) sensors, drones, etc. Traditionally, these models are trained in a centralized setup by aggregating data from edge devices to the cloud, raising privacy concerns and incurring significant communication costs. To address these challenges, there has been considerable effort towards developing on-device learning algorithms Konečný et al. (2016); Agarwal & Duchi (2011). Among these, federated learning has emerged as a popular paradigm in which numerous clients collaboratively train a global model by sharing locally computed updates with a central server. However, such setup introduces a single point of failure and demands high network bandwidth for client-server communication Assran et al. (2019). These potential issues have spurred an interest in decentralized learning, where clients are connected via a sparse topology and train through peer-to-peer communication without relying on a server. Decentralized Parallel Stochastic Gradient Descent (DPSGD) Lian et al. (2017) combines SGD with gossip averaging Xiao & Boyd (2004) and demonstrates that decentralized algorithms can achieve convergence rates comparable

---

[1]The PyTorch implementation can be found at `https://github.com/Sakshi09Ch/GFlat`

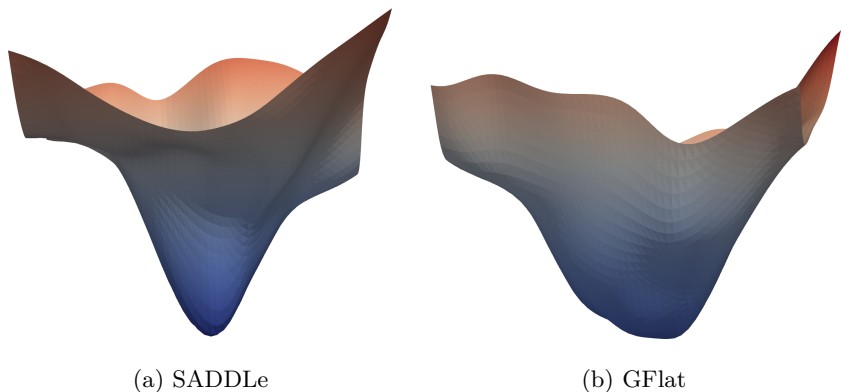

(a) SADDLe    (b) GFlat

Figure 1: Loss landscape visualization for (a) SADDLe, which uses only local perturbations, and (b) GFlat, our proposed algorithm that combines local and approximate global perturbations to achieve flatter loss landscapes in decentralized learning. Results are shown for the CIFAR-10 dataset with ResNet-20, distributed across 10 clients with extreme data heterogeneity.

to their centralized counterparts. Recently, decentralized training has shown promise in harnessing globally distributed compute resources to train large language models (LLMs) Jaghouar et al. (2024).

To achieve state-of-the-art performance, most existing decentralized learning algorithms assume data to be independently and identically distributed (IID) across clients Lian et al. (2017). However, this assumption rarely holds in real-world scenarios, where data distributions are often heterogeneous or non-IID Hsieh et al. (2020b). Understanding and mitigating the impact of non-IID data in peer-to-peer decentralized learning remains an active area of research Lin et al. (2021); Aketi et al. (2023b); Esfandiari et al. (2021); Aketi et al. (2023a); Tang et al. (2018); Choudhary et al. (2025); Koloskova et al. (2021); Pu & Nedić (2018); Takezawa et al. (2023); Vogels et al. (2021). Some approaches aim to improve model consistency by tracking global information across clients Lin et al. (2021); Aketi et al. (2023a); Koloskova et al. (2021); Takezawa et al. (2023), while others enhance local gradients by sharing cross-gradients through additional communication rounds Aketi et al. (2023b); Esfandiari et al. (2021). Recent methods attribute poor performance under non-IID settings to local overfitting and propose to seek flatter loss landscapes at each client to indirectly improve global model generalization Choudhary et al. (2025); Shi et al. (2023). However, locally flat loss landscapes do not necessarily imply global flatness, which is ultimately what matters for global model generalization and overall performance.

In this work, we alleviate the discrepancy between the flatness of the local and global objective by utilizing global information while seeking flatter loss landscapes at each client. Unlike prior approaches that rely solely on locally computed perturbations to estimate sharpness Choudhary et al. (2025); Shi et al. (2023), we propose GFlat, which incorporates an approximation of the global update direction without incurring any computational or communication overhead. Specifically, each client approximates the global perturbation by computing the difference between its current and previous model parameters. This simple yet effective strategy enables clients to estimate global sharpness, resulting in a globally flatter loss landscape compared to relying on local perturbations alone, as illustrated in Figure 1. To quantify this improvement, we track the ratio of the largest to the fifth largest Hessian eigenvalue ($\lambda_{max}/\lambda_5$), a known proxy for loss curvature Foret et al. (2021); Jastrzebski et al. (2020); Golmant et al. (2018), across training epochs. As shown in Figure 2, GFlat consistently achieves a flatter global loss surface compared to SADDLe Choudhary et al. (2025), which uses only local perturbations. Since GFlat modifies only the local optimization step, it can be seamlessly integrated into most decentralized learning frameworks without altering their communication or aggregation protocols. To demonstrate this, we also present Q-GFlat, which integrates a Quasi-Global Momentum (QGM) buffer Lin et al. (2021) to further boost performance under data heterogeneity. We present a detailed convergence analysis showing that GFlat achieves convergence rates consistent with state-of-the-art decentralized algorithms. Comprehensive experiments across multiple datasets, models, and graph topologies validate the effectiveness of our approach.



Figure 2: Ratio of largest to 5th largest eigenvalue ($\lambda_{max}/\lambda_5$) of the Hessian of the global averaged model at 4 different stages of training for SADDLe and GFlat with extreme data heterogeneity.

In summary, we make the following contributions:

- We introduce GFlat, a decentralized learning algorithm that achieves global flatness under heterogeneous data by injecting locally approximated global information into each client's optimization process, thereby enhancing generalization without extensive compute or communication overhead.

- We provide a theoretical analysis establishing convergence of GFlat to a first-order stationary point.

- Through extensive experiments on various datasets, model architectures, and graph topologies, we demonstrate that GFlat results in up to 6.75% better test accuracy compared to current state-of-the-art techniques with varying degrees of data heterogeneity.

## 2 Related Work

### 2.1 Data Heterogeneity in Decentralized Learning

Several algorithms have been proposed to address the challenge of non-IID data in decentralized learning Lin et al. (2021); Aketi et al. (2023b); Esfandiari et al. (2021); Aketi et al. (2023a); Tang et al. (2018); Choudhary et al. (2025); Koloskova et al. (2021); Vogels et al. (2021); Aketi et al. (2024). Tracking mechanisms such as Gradient Tracking Koloskova et al. (2021); Pu & Nedić (2018), Momentum Tracking Takezawa et al. (2023), and Gradient Update Tracking Aketi et al. (2023a) track global gradients or model updates to minimize the variation in local gradients across all clients. Cross-Gradient Aggregation (CGA) Esfandiari et al. (2021) and Neighborhood Gradient Mean Aketi et al. (2023b) exchange cross-gradients to align local updates between clients. However, most of the above mentioned techniques require an additional communication round. Without incurring any communication overhead, another way to improve performance with non-IID data is to synchronize the momentum buffer at each client through Quasi-Global Momentum (QGM) Lin et al. (2021). However, in the presence of extreme heterogeneity, QGM does not lead to considerable performance improvements Aketi et al. (2022). $D^2$ Tang et al. (2018) is shown to be agnostic to data heterogeneity, but its convergence requires specific constraints on the connectivity between clients. In a different vein, SADDLe Choudhary et al. (2025) replaces the local SGD optimizer with Sharpness-Aware Minimization (SAM) Foret et al. (2021) to consistently seek locally flat loss landscapes. Although SADDLe leads to impressive improvements in test accuracy, it influences global flatness in an indirect manner (refer to Figure 3), thereby leaving room for further improvements.

### 2.2 Sharpness-Aware Minimization

To improve model generalization, the authors in Foret et al. (2021) proposed Sharpness-Aware Minimization (SAM), an optimizer that simultaneously minimizes loss value as well as sharpness during training. Note that the connection between a flatter loss landscape and better generalization has been a well-studied phenomenon in deep learning Keskar et al. (2017); Izmailov et al. (2018). To further improve the performance and compute-efficiency of SAM, several variants have been proposed in the literature Kwon et al. (2021); Liu et al. (2022); Du et al. (2022); Zhao et al. (2022); Mi et al. (2022); Li et al. (2024); Wu et al. (2024);

Luo et al. (2024). Another line of research strives to shed light on the theoretical understanding of SAM by studying its convergence properties Andriushchenko & Flammarion (2022); Si & Yun (2023); Zhang et al. (2024); Oikonomou & Loizou (2025); Khanh et al. (2024). Furthermore, SAM has been shown to improve generalization in federated learning settings Dai et al. (2023); Qu et al. (2022); Caldarola et al. (2022); Sun et al. (2023); Fan et al. (2024); Lee & Yoon (2024); Caldarola et al. (2025). However, the role of sharpness in decentralized learning remains relatively unexplored. A recent theoretical result establishes that Decentralized SGD is asymptotically equivalent to average-direction SAM, suggesting a deeper connection between sharpness and decentralized optimization Zhu et al. (2023). Some recent methods have applied SAM directly to encourage local flatness in decentralized settings Choudhary et al. (2025); Chen et al. (2024); Shi et al. (2023), yet they do not explicitly consider the mismatch between local and global sharpness. To the best of our knowledge, this is the first work to explicitly identify and address the flatness discrepancy between local and global objectives in decentralized learning with heterogeneous data.

## 3 Background

In this section, we describe the decentralized peer-to-peer learning setup, the flatness-seeking optimizer SAM Foret et al. (2021), and discuss its implications towards improving generalization under data heterogeneity.

In decentralized learning, the goal is to learn a global model by aggregating models trained on locally available data at $n$ clients connected in a sparse graph topology modeled as $G = ([n], \mathbf{W})$, where $\mathbf{W}$ is the mixing matrix signifying the graph's connectivity. We assume that $G$ is strongly connected i.e., there is a path between each pair of clients Lian et al. (2017); Lin et al. (2021). Each entry $w_{ij}$ in $\mathbf{W}$ implies the effect of client $j$ on client $i$, and $w_{ij} = 0$ indicates that the $j$ and $i$ are not connected directly. The knowledge extracted from such private data is shared among peers to minimize the global loss function $f(\mathbf{x})$:

$$\min_{\mathbf{x} \in \mathbb{R}^d} f(\mathbf{x}) = \frac{1}{n} \sum_{i=1}^{n} f_i(\mathbf{x}) \tag{1}$$

Here, $f_i(\mathbf{x})$ is the local loss function at client $i$. This optimization problem is tackled by combining Stochastic Gradient Descent (SGD) with consensus-based gossip averaging Xiao & Boyd (2004).

Traditional decentralized algorithms like Decentralized Parallel Stochastic Gradient Descent (DPSGD) Lian et al. (2017) assume the data across clients to be distributed in an independent and identical manner (i.e., IID). In particular, each client $i$ in DPSGD maintains model parameters $\mathbf{x}_i^t$, computes local gradient $\mathbf{g}_i^t$ through SGD over data $\mathcal{B}_i^t \in D_i$, and incorporates neighborhood information as shown in the following update rule:

$$\mathbf{x}_i^{t+1} = \sum_{j \in \mathcal{N}(i)} w_{ij} \mathbf{x}_j^t - \eta \mathbf{g}_j^t; \quad \mathbf{g}_j^t = \nabla f_j(\mathbf{x}_j^t, \mathcal{B}_j^t) \tag{2}$$

In this work, we focus on non-IID/heterogeneous data in the form of skewed label partition, which is more closely aligned with practical learning scenarios Hsieh et al. (2020b). DPSGD performs poorly with non-IID data due to model overfitting and huge variations in local gradients across clients.

An effective approach to improving decentralized learning under non-IID data is to encourage flatter loss landscapes at each client. Prior work has established a strong correlation between model generalization and the curvature of the loss landscape, i.e., models converging to flatter minima tend to generalize better Keskar et al. (2017); Izmailov et al. (2018). Inspired by this, SADDLe Choudhary et al. (2025) utilizes a flatness-seeking optimizer called Sharpness-Aware Minimization (SAM) at each client. Originally proposed for centralized learning, SAM enhances generalization by minimizing both the loss value and its sharpness through gradient perturbations Foret et al. (2021). SAM aims to solve the following optimization problem:

$$\min_{\mathbf{x} \in \mathbb{R}^d} \{ f_\rho(\mathbf{x}) = \max_{\|\xi\| \le \rho} f(\mathbf{x} + \xi) \}, \tag{3}$$

where $\rho$ is the perturbation radius, and $\|.\|$ denotes the L2-norm. The perturbation radius $\rho$ controls the size of the neighborhood over which the loss is maximized, guiding the optimizer to converge to flatter regions.

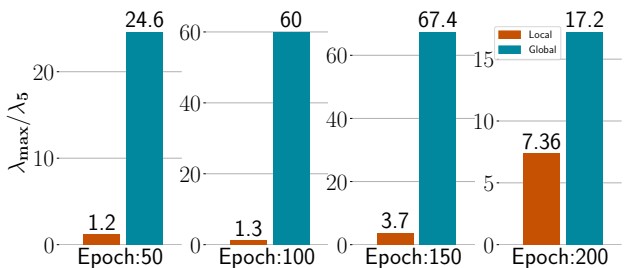

Figure 3: Ratio of largest to 5th largest eigenvalue ($\lambda_{max}/\lambda_5$) of the Hessian of the local and the global averaged model at 4 different stages of training for SADDLe.

In practice, this min-max objective is approximated via a first-order Taylor expansion of $f$, resulting in the perturbation $\xi$ being the direction of the gradient Foret et al. (2021). The final gradient is computed by minimizing the perturbed loss function $f_\rho(\mathbf{x})$ as follows:

$$\nabla f_\rho(\mathbf{x}) \approx \nabla f(\mathbf{x})|_{\mathbf{x}+\rho\cdot\frac{\mathbf{g}}{\|\mathbf{g}\|}}, \text{where } \mathbf{g} = \nabla f(\mathbf{x}) \tag{4}$$

SADDLe solves the min-max optimization shown in Equation 3 for the local training at each client. This results in a locally flat model, thus indirectly improving the generalization performance of the aggregated global model. However, due to non-IID data, the local objectives can be very different from the global objective (Equation 1). Therefore, merely focusing on minimizing local sharpness cannot effectively lead to a globally flat model. To demonstrate this, we present the local and global sharpness for SADDLe across training epochs for CIFAR-10 distributed with extreme data heterogeneity across 10 clients in Figure 3. We use the ratio of the largest to fifth-largest eigenvalue of the Hessian ($\lambda_{max}/\lambda_5$) as a proxy for sharpness. The lower the $\lambda_{max}/\lambda_5$ ratio, the flatter the loss landscape Jastrzebski et al. (2020); Foret et al. (2021); Golmant et al. (2018). As shown, SADDLe leads to lower $\lambda_{max}/\lambda_5$ for the local objective, but this does not directly translate to a low $\lambda_{max}/\lambda_5$ for the global model. To rectify this inconsistency, we propose GFlat, a novel sharpness-aware decentralized algorithm that incorporates an approximated global update into the locally computed SAM perturbation, thereby directly minimizing global sharpness and improving performance under severe data heterogeneity.

## 4 Methodology

As mentioned earlier, our goal is to minimize the sharpness of the global model in a decentralized learning setup. To achieve this, we first define the objective for global sharpness-aware minimization in decentralized learning as follows:

$$\min_{\mathbf{x}\in\mathbb{R}^d} \left\{ \max_{\|\boldsymbol{\xi}\|\leq\rho} \left[ f(\mathbf{x}+\boldsymbol{\xi}) = \frac{1}{n}\sum_{i=1}^{n} f_i(\mathbf{x}+\boldsymbol{\xi}) \right] \right\}. \tag{5}$$

Here, $\xi = \rho\frac{\nabla f(\mathbf{x})}{\|\nabla f(\mathbf{x})\|}$ is the global perturbation, approximated via a gradient ascent step scaled by the perturbation radius $\rho$. In a peer-to-peer learning scenario, data privacy concerns and communication constraints prohibit the computation of global gradient $\nabla f(\mathbf{x})$ to estimate $\xi$.

Consequently, gradient computations remain local to each client, relying solely on their respective datasets. Utilizing only these local perturbations leads to a flatter local model, but does not necessitate a flatter global model (as shown in Figure 3). One way to circumvent this challenge is to estimate the global update direction locally at each client. We propose to approximate the global gradient through model differences across two consecutive training iterations. Note that such an estimation technique has also been employed in Quasi-Global Momentum (QGM) to modify the local momentum to reduce the model discrepancy arising due to heterogeneous data Lin et al. (2021). Based on this approximation, the perturbation $\xi_i^t$ at each client

is redefined as the combination of local and global perturbations as follows:

$$\xi_i^t = \rho\Big(s.\frac{\mathbf{d}_i^t}{\|\mathbf{d}_i^t\|} + (1-s).\frac{\mathbf{g}_i^t}{\|\mathbf{g}_i^t\|}\Big), \text{where } \mathbf{d}_i^t = \mathbf{x}_i^{t-1} - \mathbf{x}_i^t \tag{6}$$

---

**Algorithm 1** GFlat: Achieving Global Flatness in Decentralized Learning with Heterogeneous Data

---

**Input:** Each client $i$ initializes model parameters $\mathbf{x}_i$, step size $\eta$, perturbation radius $\rho$, scaling factor $s \in [0,1]$, mixing matrix $\mathbf{W} = [w_{ij}]_{i,j\in[1,n]}$, $\mathcal{N}(i)$ represents neighbors of $i$ including itself.

**procedure** TRAIN( ) for $\forall i$
1.     **for** t $= 1, 2, \ldots, T$ **do**
2.         $\mathbf{g}_i^t = \nabla f_i(\mathbf{x}_i^t; \mathcal{B}_i^t)$ for $\mathcal{B}_i^t \in D_i$
3.         $\mathbf{d}_i^t = \mathbf{x}_i^{t-1} - \mathbf{x}_i^t$
4.         $\xi_i^t = \rho\big(s.\frac{\mathbf{d}_i^t}{\|\mathbf{d}_i^t\|} + (1-s).\frac{\mathbf{g}_i^t}{\|\mathbf{g}_i^t\|}\big)$
5.         $\widetilde{\mathbf{g}}_i^t = \nabla f_i(\mathbf{x}_i^t + \xi_i^t; \mathcal{B}_i^t)$
6.         $\mathbf{x}_i^{(t+1/2)} = \mathbf{x}_i^{(t)} - \eta\widetilde{\mathbf{g}}_i^t$
7.         SENDRECEIVE($\mathbf{x}_i^{(t+1/2)}$)
8.         $\mathbf{x}_i^{t+1} = \sum_{j\in\mathcal{N}_i^{(t)}} w_{ij}\mathbf{x}_j^{(t+1/2)}$
9.     **end**
**return** $\bar{\mathbf{x}}^T = \frac{1}{n}\sum_{i=1}^n \mathbf{x}_i^T$

---

Here, $s$ is a scaling factor that defines the emphasis placed on global vs local perturbation. Intuitively, in the presence of extreme data heterogeneity, more value should be given to the global perturbation $\mathbf{d}_i^t/\|\mathbf{d}_i^t\|$, as the local gradients overfit the locally available data. Similarly, for milder forms of non-IIDness, it may suffice to emphasize the local perturbation $\mathbf{g}_i^t/\|\mathbf{g}_i^t\|$ over its global counterpart. We observe this in our experiments, where $s$ remains closer to 1 for extreme heterogeneity (i.e., $\alpha = 0.001$). We provide additional discussion on local and global flatness discrepancy in Appendix E.

Algorithm 1 presents the pseudocode of GFlat. At each training iteration $t$, GFlat computes the stochastic gradient $\mathbf{g}_i^t$ at each client $i$. Then it estimates the global perturbation through model differences $(\mathbf{x}_i^{t-1} - \mathbf{x}_i^t)$ as shown in line 3, Algorithm 1. The SAM perturbation $\xi_i^t$ is calculated as a weighted combination of local and global perturbations (line 4, Algorithm 1). All clients calculate $\widetilde{\mathbf{g}}_i^t$ for the perturbed model $(\mathbf{x}_i^t + \xi_i^t)$ to perform local updates (line 6). The clients then exchange models with their peers and perform gossip averaging (line 8) Lian et al. (2017). This continues for a predefined set of $T$ iterations, and we evaluate the consensus model $\bar{\mathbf{x}}^T$ Lian et al. (2017); Aketi et al. (2023a). Note that computing $\mathbf{d}_i^t$ to estimate the global perturbation requires minimal compute, incurs no communication overhead, and $\mathcal{O}(m)$ memory at each client, where $m$ denotes the number of trainable model parameters. In essence, GFlat approximates the global update direction and incorporates this in local perturbation at each client to target global sharpness without any communication overhead.

As GFlat modifies the local optimizer, it is effectively complementary to existing decentralized algorithms for tackling data heterogeneity and can be used in synergy with them. To that effect, we present a version of GFlat termed Q-GFlat, which incorporates a Quasi-Global Momentum (QGM) buffer Lin et al. (2021) to further reduce the model inconsistency. While QGM mimics the global update direction through a modified local momentum, GFlat focuses on seeking a flatter minimum for the global model by injecting global information in the local perturbation (i.e., ascent step) for the SAM optimizer. Together, these techniques greatly improve the performance in the presence of non-IID data. Please refer to Appendix C for implementation details.

## 5 Convergence Rate Analysis

This section provides the convergence analysis for the proposed GFlat algorithm for a general non-convex loss objective. We state the following standard assumptions:

**Assumption 1** *Lipschitz Gradients*: Each function $f_i(\mathbf{x})$ is L-smooth i.e., $||\nabla f_i(\mathbf{y}) - \nabla f_i(\mathbf{x})|| \leq L||\mathbf{y} - \mathbf{x}|| \quad \forall \mathbf{x}, \mathbf{y}$.

**Assumption 2** *Bounded Variance*: The variance of the stochastic gradients is assumed to be bounded. There exist constants $\sigma$ and $\delta$ such that

$$\mathbb{E}_{\mathcal{B} \sim \mathcal{D}_i} \left\| \frac{\nabla f_i(\mathbf{x}; \mathcal{B})}{\|\nabla f_i(\mathbf{x}; \mathcal{B})\|} - \frac{\nabla f_i(\mathbf{x})}{\|\nabla f_i(\mathbf{x})\|} \right\|^2 \leq \sigma^2 \quad \forall i \tag{7}$$

$$\|\nabla f_i(\mathbf{x}) - \nabla f(\mathbf{x})\|^2 \leq \delta^2 \tag{8}$$

**Assumption 3** *Doubly Stochastic Mixing Matrix*: $\mathbf{W}$ is a real doubly stochastic matrix with $\lambda_1(\mathbf{W}) = 1$ and $max\{|\lambda_2(\mathbf{W})|, |\lambda_N(\mathbf{W})|\} \leq \sqrt{\lambda} < 1$, where $\lambda_i(\mathbf{W})$ is the $i^{th}$ largest eigenvalue of $\mathbf{W}$ and $\lambda$ is a constant.

These assumptions align with those used in existing decentralized learning algorithms Lian et al. (2017); Aketi et al. (2023a;b); Esfandiari et al. (2021); Choudhary et al. (2024); Lin et al. (2021). Assumption 2 introduces a slight variation of the standard stochastic variance bound (Equation 7), motivated by recent works exploring flatter loss landscapes in federated learning settings (detailed discussion in Appendix B.1)Qu et al. (2022); Lee & Yoon (2024); Fan et al. (2024). Theorem 1 presents the convergence for our proposed GFlat algorithm (proof in Appendix B.2).

**Theorem 1** *Given Assumptions 1-3, let the learning rate satisfy* $\eta \leq \frac{\sqrt{(1-\sqrt{\lambda})^2 + 16(1-\sqrt{\lambda})} - (1-\sqrt{\lambda})}{8L}$. *Then, for all* $T \geq 1$, *we have:*

$$\frac{1}{T} \sum_{t=0}^{T-1} \mathbb{E}\left[\left\|\nabla f\left(\bar{\mathbf{x}}^t\right)\right\|^2\right] \leq \frac{2}{\eta T}\left(f(\bar{\mathbf{x}}^0) - f^\star\right) + \sigma^2 L^2 \rho^2 (1-s)^2 \left(\frac{4\eta^2 L^2}{(1-\sqrt{\lambda})^2} + \frac{\eta L}{n}\right) + \delta^2 L^2 \left(\frac{12\eta^2}{(1-\sqrt{\lambda})^2}\right) +$$
$$\rho^2 L^2 \left[\frac{8\eta^2 L^2}{(1-\sqrt{\lambda})^2}\left(2(1-s)^2 + s^2\epsilon^2 + 3\right) + 3s^2\epsilon^2 + 6s^2 + 3\right] \tag{9}$$

*where,* $\epsilon = \max_T \max_{1 \leq i \leq n} \|\frac{\mathbf{d}_i}{\|\mathbf{d}_i\|} - \frac{\nabla f(\mathbf{x_i})}{\|\nabla f(\mathbf{x_i})\|}\|$ *denotes the worst case error in approximating the global update direction for the GFlat perturbation* $\xi_i^t$.

The result of Theorem 1 shows that the averaged gradient of the consensus (averaged) model $\bar{\mathbf{x}}^t$ is upper-bounded by the sub-optimality gap $f(\bar{\mathbf{x}}^0) - f^\star$, the stochastic variance $\sigma$, global variance $\delta$ due to data heterogeneity, the perturbation radius $\rho$ and the approximation error $\epsilon$. Further, we present a corollary to demonstrate the convergence rate of GFlat in terms of training iterations $T$. Please refer to Appendix B.4 for the detailed proof.

**Corollary 2** *Suppose that the step size satisfies* $\eta = \mathcal{O}\left(\sqrt{\frac{n}{T}}\right)$ *and the perturbation radius* $\rho = \mathcal{O}\left(\sqrt{\frac{1}{T}}\right)$. *For sufficiently large* $T$ *we have,*

$$\frac{1}{T} \sum_{t=0}^{T-1} \mathbb{E}\left[\left\|\nabla f\left(\bar{\mathbf{x}}^t\right)\right\|^2\right] \leq \mathcal{O}\left(\frac{f(\bar{\mathbf{x}}^0) - f^\star}{\sqrt{nT}} + \frac{(1-s)\sigma^2}{T^{3/2}} + \frac{\delta^2}{T} + \frac{\epsilon^2}{T}\right) \tag{10}$$

The dominant term in Corollary 2 is $(1/\sqrt{nT})$, which determines the overall convergence rate of GFlat. This matches the standard convergence behavior observed in decentralized algorithms, as established in existing literature Lian et al. (2017); Lin et al. (2021); Aketi et al. (2023a); Esfandiari et al. (2021).

**Remark 1.** A closer inspection of Equation 10 reveals that the influence of stochastic variance ($\sigma$) is modulated by the factor $(1-s)$. In essence, for higher $s$, GFlat can speed up the convergence by alleviating the impact of stochastic variance. As a result, higher values of $s$ (i.e., closer to 1) lead to reduced variance

contribution, thereby accelerating convergence. Through our experiments, we observe that GFlat achieves optimal performance for $0.5 <= s <= 1$, corroborating this theoretical insight.

**Remark 2.** The approximation error in estimating the global perturbation ($\epsilon$) appears as a higher-order term in Equation 10, scaled by $1/T$, and thus its impact on convergence diminishes as training progresses. Consequently, we observe performance gains reported in Tables 1–5, even when using an approximation of global perturbation (i.e., $\mathbf{d}_i^t$ in Equation 6).

# 6 Experiments

## 6.1 Experimental Setup

We conduct experiments on diverse datasets, model architectures, graph topologies, and sizes. The analysis is presented on - (a) **datasets**: CIFAR-10, CIFAR-100, Imagenette Husain (2018) and ImageNet Deng et al. (2009), (b) **models**: ResNet-20, ResNet-18 and MobileNet-v2, (c) **graph topologies**: undirected ring with 2 peers/client and torus with 4 peers/client with uniform mixing matrix (Figure 7), and (d) **graph sizes**: 10 to 40 clients. We generate disjoint non-IID data partitions across clients using a Dirichlet distribution Hsu et al. (2019), where the class proportions for each client are drawn from $\mathrm{Dir}(\alpha)$. A smaller concentration parameter $\alpha$ yields more skewed class distributions, where each client receives data from fewer dominant classes, resulting in higher statistical heterogeneity (see Figure 6). The partitions are fixed at initialization and remain unchanged during training. We compare GFlat with DPSGD Lian et al. (2017) and SADDLe Choudhary et al. (2025), and the Q-GFlat variant with QGM Lin et al. (2021) and Q-SADDLe Choudhary et al. (2025). We report the test accuracy of the consensus model $\bar{\mathbf{x}}^T$ averaged over three randomly chosen seeds. Please refer to Appendix D.4 for details related to training hyperparameters.

## 6.2 Results

Table 1: Test accuracy of various decentralized algorithms evaluated on CIFAR-10 and Imagenette distributed with different degrees of heterogeneity for various models over a ring topology. We also include results over the IID baseline, which serves as an upper bound on performance.

| Clients | Method | CIFAR-10 (ResNet20) | | Imagenette (Mobilenet-v2) | |
|---|---|---|---|---|---|
| | | $\alpha = 0.01$ | $\alpha = 0.001$ | $\alpha = 0.01$ | $\alpha = 0.001$ |
| 10 | DPSGD (IID) | $90.46 \pm 0.33$ | | $75.15 \pm 0.42$ | |
| | DPSGD | $49.17 \pm 17.38$ | $40.74 \pm 2.62$ | $40.11 \pm 4.85$ | $34.50 \pm 5.88$ |
| | SADDLe | $64.58 \pm 5.63$ | $61.30 \pm 0.79$ | $45.95 \pm 3.03$ | $42.37 \pm 4.30$ |
| | *GFlat (ours)* | $\mathbf{70.12 \pm 0.68}$ | $\mathbf{62.29 \pm 0.66}$ | $\mathbf{48.91 \pm 2.51}$ | $\mathbf{48.07 \pm 1.01}$ |
| 20 | DPSGD (IID) | $89.46 \pm 0.02$ | | $73.25 \pm 0.49$ | |
| | DPSGD | $40.49 \pm 3.06$ | $36.13 \pm 5.67$ | $36.45 \pm 0.46$ | $30.96 \pm 1.55$ |
| | SADDLe | $52.14 \pm 2.02$ | $47.06 \pm 2.25$ | $39.18 \pm 1.04$ | $37.72 \pm 1.16$ |
| | *GFlat (ours)* | $\mathbf{56.21 \pm 1.33}$ | $\mathbf{51.76 \pm 0.68}$ | $\mathbf{39.95 \pm 0.51}$ | $\mathbf{39.23 \pm 1.26}$ |

**Performance Comparison:** Table 1 presents results for GFlat across CIFAR-10 and Imagenette under two different levels of heterogeneity (with lower $\alpha$ indicating higher non-IIDness). As shown, GFlat achieves a 1-5.5% improvement in accuracy over SADDLe, highlighting the benefits of incorporating global perturbation into the local SAM step. Similarly, on Imagenette, GFlat results in up to 5.7% better accuracy compared to SADDLe Choudhary et al. (2025). We demonstrate the efficacy of Q-GFlat in Table 2, comparing it against QGM and Q-SADDLe across 10-40 clients connected in a ring topology. On CIFAR-10 and CIFAR-100, Q-GFlat outperforms Q-SADDLe by ∼0.6% and ∼2.6%, respectively. We present additional results on Imagenette in Table 3, where Q-GFlat yields an average improvement of 3.8% for 10-20 clients in a ring topology. To study the impact of graph topology, results on a 20-client torus topology are reported in Table 4. Both GFlat and Q-GFlat demonstrate significant gains over their respective baselines, SADDLe and Q-SADDLe, achieving an average improvement of 2.2% for $\alpha = 0.01, 0.001$.

Table 2: Test accuracy of various decentralized algorithms evaluated on CIFAR-10 and CIFAR-100 distributed with different degrees of heterogeneity over ResNet-20 over a ring topology.

| Clients | Method | CIFAR-10 | | CIFAR-100 | |
|---|---|---|---|---|---|
| | | $\alpha = 0.01$ | $\alpha = 0.001$ | $\alpha = 0.01$ | $\alpha = 0.001$ |
| 10 | QGM | $77.41 \pm 8.00$ | $79.48 \pm 2.76$ | $48.06 \pm 4.36$ | $44.16 \pm 6.71$ |
| | Q-SADDLe | $87.72 \pm 1.59$ | $86.33 \pm 0.24$ | $58.06 \pm 0.68$ | $56.76 \pm 0.86$ |
| | *Q-GFlat (ours)* | $\mathbf{88.02 \pm 1.04}$ | $\mathbf{86.73 \pm 0.20}$ | $\mathbf{58.98 \pm 0.33}$ | $\mathbf{58.12 \pm 0.44}$ |
| 20 | QGM | $72.20 \pm 0.77$ | $62.48 \pm 8.56$ | $45.23 \pm 3.26$ | $44.48 \pm 4.53$ |
| | Q-SADDLe | $84.17 \pm 0.73$ | $82.81 \pm 0.89$ | $52.59 \pm 0.48$ | $48.20 \pm 0.93$ |
| | *Q-GFlat (ours)* | $\mathbf{84.61 \pm 0.70}$ | $\mathbf{83.04 \pm 0.41}$ | $\mathbf{54.34 \pm 0.65}$ | $\mathbf{53.45 \pm 0.62}$ |
| 40 | QGM | $70.46 \pm 4.14$ | $60.86 \pm 0.98$ | $40.15 \pm 0.90$ | $38.73 \pm 1.47$ |
| | Q-SADDLe | $77.49 \pm 0.83$ | $73.54 \pm 2.04$ | $43.25 \pm 1.71$ | $41.99 \pm 1.27$ |
| | *Q-GFlat (ours)* | $\mathbf{77.97 \pm 1.01}$ | $\mathbf{75.34 \pm 1.13}$ | $\mathbf{46.37 \pm 0.79}$ | $\mathbf{45.25 \pm 0.41}$ |

Table 3: Test accuracy of various decentralized algorithms evaluated on Imagenette distributed with different degrees of heterogeneity over MobileNet-v2 over a ring topology.

| Clients | Method | Imagenette (Mobilenet-v2) | |
|---|---|---|---|
| | | $\alpha = 0.01$ | $\alpha = 0.001$ |
| 10 | QGM | $56.30 \pm 4.03$ | $45.82 \pm 5.99$ |
| | Q-SADDLe | $62.35 \pm 3.64$ | $63.18 \pm 1.59$ |
| | *Q-GFlat (ours)* | $\mathbf{64.98 \pm 0.96}$ | $\mathbf{64.89 \pm 0.56}$ |
| 20 | QGM | $52.52 \pm 4.88$ | $45.67 \pm 3.10$ |
| | Q-SADDLe | $53.64 \pm 4.60$ | $53.99 \pm 3.45$ |
| | *Q-GFlat (ours)* | $\mathbf{60.39 \pm 2.40}$ | $\mathbf{58.08 \pm 1.50}$ |

Table 4: Test accuracy of various decentralized algorithms evaluated on Imagenette distributed with different degrees of heterogeneity over a torus topology.

| Clients | Method | Imagenette (Mobilenet-v2) | |
|---|---|---|---|
| | | $\alpha = 0.01$ | $\alpha = 0.001$ |
| 20 | DPSGD | $35.75 \pm 1.91$ | $27.40 \pm 1.19$ |
| | SADDLe | $\mathbf{40.55 \pm 4.19}$ | $33.23 \pm 0.27$ |
| | *GFlat (ours)* | $40.11 \pm 2.00$ | $\mathbf{34.63 \pm 0.83}$ |
| | QGM | $48.28 \pm 5.88$ | $42.46 \pm 10.75$ |
| | Q-SADDLe | $58.68 \pm 3.04$ | $54.78 \pm 5.81$ |
| | *Q-GFlat (ours)* | $\mathbf{61.98 \pm 1.17}$ | $\mathbf{59.19 \pm 0.39}$ |

Across all experiments, we observe that GFlat exhibits greater robustness to random seed variations compared to the baselines, with significantly lower variance in performance. To further emphasize the effectiveness of our approach, we conduct experiments on ImageNet using the ResNet-18 architecture, distributed across 10 clients with three different levels of data heterogeneity. As shown in Table 5, GFlat outperforms SADDLe by 3% on average.

**Impact of Scaling Factor:** To evaluate the impact of the scaling factor $s$, we conduct an ablation study using the CIFAR-10 dataset trained with a ResNet-20 architecture distributed across 10 clients under a highly non-IID setting ($\alpha = 0.01$). As shown in Figure 5, $s \in \{0.5, 0.7, 1\}$ consistently yields higher test accuracy compared to $s \in \{0, 0.2\}$, suggesting that

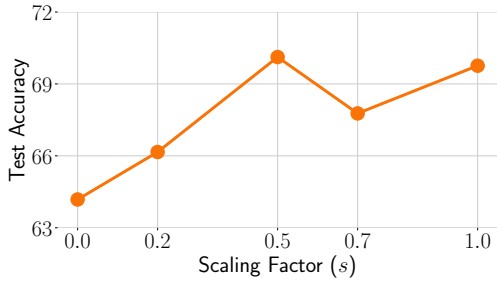

Figure 5: Impact of scaling factor $s$ on performance for CIFAR-10 distributed across 10 clients with $\alpha = 0.01$.

Table 5: Test accuracy of various decentralized algorithms evaluated on Imagenet distributed with different degrees of heterogeneity ($\alpha$) over a ring topology.

| Clients | Method | Imagenet (ResNet-18) | | |
|---|---|---|---|---|
| | | $\alpha = 0.1$ | $\alpha = 0.01$ | $\alpha = 0.001$ |
| 10 | DPSGD (IID) | | 66.73 | |
| | DPSGD | 50.00 | 45.49 | 46.77 |
| | SADDLe | 54.63 | 51.28 | 51.26 |
| | *GFlat (ours)* | **57.54** | **54.49** | **52.40** |

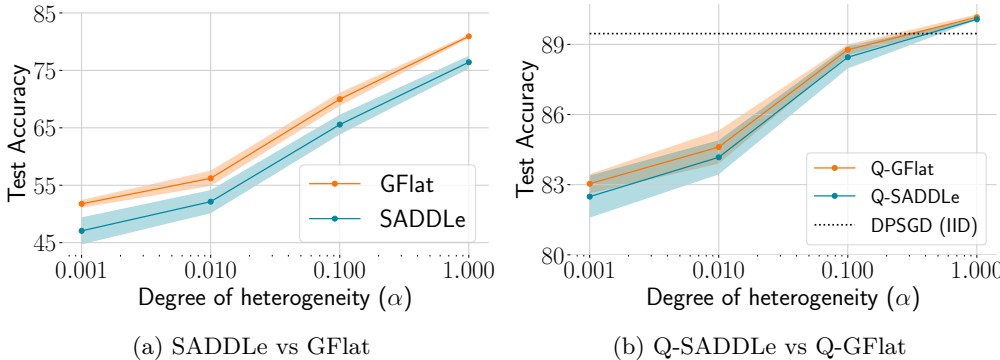

(a) SADDLe vs GFlat          (b) Q-SADDLe vs Q-GFlat

Figure 4: Test accuracy for the CIFAR-10 dataset trained on ResNet-20 architecture distributed with varying degrees of data heterogeneity over 20 clients.

larger emphasis on the global direction improves performance. This aligns with the insights gained from our theoretical analysis (Remark 1).

**Robustness to Degrees of Data Heterogeneity:** We perform an ablation study to analyze the impact of data heterogeneity on the performance of GFlat. Figure 4 presents the test accuracy of GFlat and Q-GFlat across different values of $\alpha$, where lower $\alpha$ indicates higher non-IIDness. As observed, both variants consistently outperform their respective baselines. Notably, Q-GFlat and Q-SADDLe even surpass the performance of DPSGD under IID data for $\alpha = 1$, highlighting the effectiveness of our approach.

## 7   Limitations and Future Work

GFlat introduces a scaling factor $s$ to balance the relative emphasis on local versus global perturbations. We keep $s$ fixed throughout the training and tune it as a hyperparameter. However, since the alignment between local and global models can vary over training iterations, $s$ could be made adaptive. For instance, clients may benefit from a higher $s$ when the divergence between local and gossip-averaged models is large, and a lower value when they are well aligned. We leave the design and evaluation of such an adaptive strategy to future work. Another promising direction for future work is the integration of computationally efficient variants of SAM Du et al. (2022); Liu et al. (2022) into our framework, followed by an analysis of their impact on performance.

## 8   Conclusion

In this work, we highlight that under severe data heterogeneity in decentralized training, locally flat loss landscapes at individual clients do not necessarily translate into flatness of the global loss objective. To address this mismatch, we propose GFlat, a novel algorithm that achieves globally flat loss landscapes by incorporating an approximated global direction into the local optimization process. We provide a theoretical analysis to establish that GFlat achieves a convergence rate of $\mathcal{O}(1/\sqrt{nT})$, similar to state-of-the-art decentralized algorithms. Furthermore, we validate our approach through extensive experiments conducted

across diverse datasets, model architectures, and graph topologies. Our results show that GFlat improves test accuracy by up to 6.75% over current state-of-the-art decentralized algorithms without any additional communication cost.

## 9 Acknowledgments

The authors would like to thank Utkarsh Saxena for insightful technical discussions. This work was supported in part by, the Center for the Co-Design of Cognitive Systems (COCOSYS), a DARPA-sponsored JUMP center, the Semiconductor Research Corporation (SRC) and the National Science Foundation (NSF).

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

## A Appendix

## B Theoretical Analysis

### B.1 Discussion on Stochastic Variance Assumption

As mentioned in Section 5, we assume a slight variation of the following standard stochastic variance bound presented in contemporary decentralized learning works:

$$\mathbb{E}_{\mathcal{B}\sim\mathcal{D}_i}||\nabla f_i(\mathbf{x};\mathcal{B}) - \nabla f_i(\mathbf{x})||^2 \leq \sigma^2 \tag{11}$$

We rewrite our assumption here (Equation 7) for ease of understanding:

$$\mathbb{E}_{\mathcal{B}\sim\mathcal{D}_i}\left\|\frac{\nabla f_i(\mathbf{x};\mathcal{B})}{\|\nabla f_i(\mathbf{x};\mathcal{B})\|} - \frac{\nabla f_i(\mathbf{x})}{\|\nabla f_i(\mathbf{x})\|}\right\|^2 \leq \sigma^2 \ \forall i \tag{12}$$

Concretely, instead of bounding the norm of the difference between $\nabla f_i(\mathbf{x};\mathcal{B})$ and the true gradient $\nabla f_i(\mathbf{x})$ for each client, as shown in Equation 11, we bound the difference in their directions Lee & Yoon (2024); Qu et al. (2022). We argue that the bound in Equation 11 may be loose and less informative in practice Qu et al. (2022), particularly during the later stages of training, when the gradient norms are small. In such cases, even a small variance in magnitude can lead to significant directional variance, potentially causing instability in optimization. This distinction is crucial, as the convergence of the consensus model depends not only on the magnitude of the gradients at each client but also on their alignment with the true update directions. Moreover, $\frac{\nabla f_i(\mathbf{x};\mathcal{B})}{\|\nabla f_i(\mathbf{x};\mathcal{B})\|}$ and $\frac{\nabla f_i(\mathbf{x})}{\|\nabla f_i(\mathbf{x})\|}$ are unit vectors, and thus their euclidean distance can be reasonably bounded by the angle between them.

### B.2 Convergence Rate Proof

In this work, we solve the optimization problem of minimizing the global loss function $f(x)$ distributed across $n$ clients:

$$\min_{\mathbf{x}\in\mathbb{R}^d} f(\mathbf{x}) = \frac{1}{n}\sum_{i=1}^{n} f_i(\mathbf{x}), \text{where } f_i(\mathbf{x}) = \mathbb{E}_{\mathcal{B}_i^t\in D_i}[f_i(\mathbf{x};\mathcal{B}_i^t)] \ \forall i \tag{13}$$

We begin by providing an upper bound on the consensus error, defined as $\frac{1}{n}\sum_{i=1}^{n}\|\mathbf{x}_i^t - \bar{\mathbf{x}}^t\|^2$, which quantifies the average deviation of local models $\mathbf{x}_i^t$ from the global averaged model $\bar{\mathbf{x}}^t$ at training iteration $t$. The consensus error reflects the effectiveness of gossip-based averaging in decentralized learning and indicates how closely clients are approaching global agreement.

**Lemma 3** *Given assumptions 1-3, the distance between the average sequence iterate $\bar{\mathbf{x}}^t$ and the sequence iterates $\mathbf{x}_i^t$ (i.e. the consensus error) is given by*

$$\sum_{t=1}^{T-1}\frac{1}{n}\mathbb{E}\left[\sum_{i=1}^{n}\|\mathbf{x}_i^t - \bar{\mathbf{x}}^t\|^2\right] \leq \frac{4\eta^2 T}{(1-\sqrt{\lambda})^2}\left(L^2\rho^2(1-s)^2\sigma^2 + 3\delta^2 + 4L^2(1-s)^2\rho^2 + \right.$$
$$\left. 2L^2 s^2\epsilon^2\rho^2 + 6L^2\rho^2\right) + \frac{4\eta^2}{(1-\sqrt{\lambda})}\sum_{t=0}^{T-1}\mathbb{E}[\|\frac{1}{n}\sum_{i=1}^{n}\nabla f_i(\mathbf{x}_i^t + \xi_i^t)\|^2] \tag{14}$$

*Proof:* We introduce the following notations and the subsequent properties Esfandiari et al. (2021); Aketi et al. (2023b); Choudhary et al. (2024):

$$
\begin{aligned}
\mathbf{Q} &= \frac{1}{n}\mathbf{1}\mathbf{1}^\top \\
\widetilde{\mathbf{G}}^t &\triangleq [\widetilde{\mathbf{g}}_1^t, \widetilde{\mathbf{g}}_2^t, ..., \widetilde{\mathbf{g}}_n^t] \\
\mathbf{X}^t &\triangleq [\mathbf{x}_1^t, \mathbf{x}_2^t, ..., \mathbf{x}_n^t] \\
\mathbf{G}^t &\triangleq [\mathbf{g}_1^t, \mathbf{g}_2^t, ..., \mathbf{g}_n^t] \\
\mathbf{H}^t &\triangleq [\nabla f_1(\mathbf{x}_1^t + \xi_1^t), \nabla f_2(\mathbf{x}_2^t + \xi_2^t), ..., \nabla f_N(\mathbf{x}_n^t + \xi_n^t)]
\end{aligned}
\tag{15}
$$

Note that $\widetilde{\mathbf{g}}_i^t$ represents the local stochastic gradient computed using GFlat as shown in Algorithm 1 (line 5). For all the above matrices, $\|\mathbf{A}\|_{\mathfrak{F}}^2 = \sum_{i=1}^n \|\mathbf{a}_i\|^2$, where $\mathbf{a}_i$ is the $i$-th column of the matrix $\mathbf{A}$. Therefore, we have:

$$
\|\mathbf{X}^t(\mathbf{I} - \mathbf{Q})\|_{\mathfrak{F}}^2 = \sum_{i=1}^n \|\mathbf{x}_i^t - \bar{\mathbf{x}}^t\|^2.
\tag{16}
$$

For each doubly stochastic matrix $\mathbf{W}$, the following properties hold true:

- $\mathbf{Q}\mathbf{W} = \mathbf{W}\mathbf{Q}$;

- $(\mathbf{I} - \mathbf{Q})\mathbf{W} = \mathbf{W}(\mathbf{I} - \mathbf{Q})$;

- For any integer $t \geq 1$, $\|(\mathbf{I} - \mathbf{Q})\mathbf{W}\|_{\mathfrak{G}} \leq (\sqrt{\lambda})^t$ ($\|\cdot\|_{\mathfrak{G}}$ is the spectrum norm of a matrix)

For $n$ arbitrary real square matrices $\mathbf{A}_i, i \in \{1, 2, ..., n\}$,

$$
\|\sum_{i=1}^n \mathbf{A}_i\|_{\mathfrak{F}}^2 \leq \sum_{i=1}^n \sum_{j=1}^n \|\mathbf{A}_i\|_{\mathfrak{F}}\|\mathbf{A}_j\|_{\mathfrak{F}}.
\tag{17}
$$

We now proceed to provide a bound on the consensus error. Since $\mathbf{X}^t = \mathbf{X}^{t-1}\mathbf{W} - \eta\widetilde{\mathbf{G}}^t$ we have:

$$
\mathbf{X}^t(\mathbf{I} - \mathbf{Q}) = \mathbf{X}^{t-1}(\mathbf{I} - \mathbf{Q})\mathbf{W} - \eta\tilde{\mathbf{G}}^t(\mathbf{I} - \mathbf{Q})
\tag{18}
$$

Applying the above equation $t$ times, we have:

$$
\mathbf{X}^t(\mathbf{I} - \mathbf{Q}) = \mathbf{X}^0(\mathbf{I} - \mathbf{Q})\mathbf{W}^t - \sum_{\tau=1}^t \eta\tilde{\mathbf{G}}^\tau(\mathbf{I} - \mathbf{Q})\mathbf{W}^{t-\tau} = -\eta\sum_{\tau=1}^t \tilde{\mathbf{G}}^\tau(\mathbf{I} - \mathbf{Q})\mathbf{W}^{t-\tau}
\tag{19}
$$

$$
\begin{aligned}
\mathbb{E}\left[\left\|\mathbf{X}^t(\mathbf{I} - \mathbf{Q})\right\|_{\mathfrak{F}}^2\right] &= \eta^2\mathbb{E}\left[\left\|\sum_{\tau=0}^{t-1} \tilde{\mathbf{G}}^\tau(\mathbf{I} - \mathbf{Q})\mathbf{W}^{t-1-\tau}\right\|_{\mathfrak{F}}^2\right] \\
&\leq \eta^2\sum_{\tau=0}^{t-1}\sum_{\tau'=0}^{t-1} \mathbb{E}\left[\left\|\tilde{\mathbf{G}}^\tau(\mathbf{I} - \mathbf{Q})\mathbf{W}^{t-1-\tau}\right\|_{\mathfrak{F}}\left\|\tilde{\mathbf{G}}^{\tau'}(\mathbf{I} - \mathbf{Q})\mathbf{W}^{t-1-\tau'}\right\|_{\mathfrak{F}}\right] \\
&\leq \eta^2\sum_{\tau=0}^{t-1}\sum_{\tau'=0}^{t-1} \lambda^{(t-1-\frac{\tau+\tau'}{2})}\mathbb{E}[\|\widetilde{\mathbf{G}}^\tau\|_{\mathfrak{F}}\|\widetilde{\mathbf{G}}^{\tau'}\|_{\mathfrak{F}}] \overset{a}{\leq} \eta^2\sum_{\tau=0}^{t-1}\sum_{\tau'=0}^{t-1} \lambda^{(t-1-\frac{\tau+\tau'}{2})}\left(\frac{1}{2}\mathbb{E}[\|\widetilde{\mathbf{G}}^\tau\|_{\mathfrak{F}}^2] + \frac{1}{2}\mathbb{E}[\|\widetilde{\mathbf{G}}^{\tau'}\|_{\mathfrak{F}}^2]\right) \\
&= \eta^2\sum_{\tau=0}^{t-1}\sum_{\tau'=0}^{t-1} \lambda^{(t-1-\frac{\tau+\tau'}{2})}\mathbb{E}[\|\widetilde{\mathbf{G}}^\tau\|_{\mathfrak{F}}^2] \overset{b}{\leq} \frac{\eta^2}{(1 - \sqrt{\lambda})}\sum_{\tau=0}^{t-1} \lambda^{\frac{(t-1-\tau)}{2}}\mathbb{E}[\|\widetilde{\mathbf{G}}^\tau\|_{\mathfrak{F}}^2]
\end{aligned}
\tag{20}
$$

(a) follows from the inequality $xy \leq \frac{1}{2}(x^2 + y^2)$ for any two real numbers $x, y$.

(b) follows from $\sum_{\tau'=0}^{t-1} \lambda^{(t-1-\frac{\tau'+\tau}{2})} \leq \frac{\lambda^{\frac{(t-1-\tau)}{2}}}{1-\sqrt{\lambda}}$.

We proceed with finding the bounds for $\mathbb{E}[\|\widetilde{\mathbf{G}}^\tau\|_{\tilde{\mathfrak{F}}}^2]$:

$$\sum_{i=1}^{n} \mathbb{E}[\|\nabla f_i(\mathbf{x}_i^\tau + (\xi_i^\tau; \mathcal{B}_i^\tau)) - \nabla f_i(\mathbf{x}_i^\tau + \xi_i^\tau) + \nabla f_i(\mathbf{x}_i^\tau + \xi_i^\tau) - \nabla f_i(\mathbf{x}_i^\tau + \xi_{true}^\tau) + \nabla f_i(\mathbf{x}_i^\tau + \xi_{true}^\tau)$$

$$- \nabla f(\mathbf{x}_i^\tau + \xi_i^\tau) + \nabla f(\mathbf{x}_i^\tau + \xi_i^\tau)\|^2] \leq 4 \sum_{i=1}^{n} \underbrace{\mathbb{E}[\|\nabla f_i(\mathbf{x}_i^\tau + (\xi_i^\tau; \mathcal{B}_i^\tau))) - \nabla f_i(\mathbf{x}_i^\tau + \xi_i^\tau)\|]^2}_{I} \tag{21}$$

$$+ \underbrace{\mathbb{E}[\|\nabla f_i(\mathbf{x}_i^\tau + \xi_i^\tau) - \nabla f_i(\mathbf{x}_i^\tau + \xi_{true}^\tau)\|]^2}_{II} + \underbrace{\mathbb{E}[\|\nabla f_i(\mathbf{x}_i^\tau + \xi_{true}^\tau) - \nabla f(\mathbf{x}_i^\tau + \xi_i^\tau)\|]^2}_{III} + \mathbb{E}[\|\nabla f(\mathbf{x}_i^\tau + \xi_i^\tau)\|]^2$$

Here, $\xi_{true}^\tau = \rho \frac{\bar{\mathbf{g}}^\tau}{\|\bar{\mathbf{g}}^\tau\|}$ is the true perturbation, where $\bar{\mathbf{g}}^\tau = \nabla f(\mathbf{x}_i^\tau) = \frac{1}{n}\sum_{i=1}^{n} \nabla f_i(\mathbf{x}_i^\tau)$ denotes the average gradient across all clients. $\epsilon_i = \|\frac{\mathbf{d}_i^\tau}{\|\mathbf{d}_i^\tau\|} - \frac{\bar{\mathbf{g}}^\tau}{\|\bar{\mathbf{g}}^\tau\|}\|$ is the error introduced due to the global update approximation computed through the model differences $\mathbf{d}_i^\tau$ and $\epsilon = \max_T \max_{1 \leq i \leq n} \epsilon_i$.

Now we bound $I$, $II$, and $III$:

$$\mathbb{E}[\|\nabla f_i(\mathbf{x}_i^\tau + (\xi_i^\tau; d_i^\tau)) - \nabla f_i(\mathbf{x}_i^\tau + \xi_i^\tau)\|^2] \leq L^2 \rho^2 \mathbb{E}\left[\left\|s \cdot \frac{\mathbf{d}_i^\tau}{\|\mathbf{d}_i^\tau\|} + (1-s) \cdot \frac{\mathbf{g}_i^\tau}{\|\mathbf{g}_i^\tau\|} - s \cdot \frac{\mathbf{d}_i^\tau}{\|\mathbf{d}_i^\tau\|} - (1-s) \cdot \frac{\nabla f_i(\mathbf{x}_i^\tau)}{\|\nabla f_i(\mathbf{x}_i^\tau)\|}\right\|^2\right]$$

$$= L^2 \rho^2 (1-s)^2 \mathbb{E}\left[\left\|\frac{\mathbf{g}_i^\tau}{\|\mathbf{g}_i^\tau\|} - \frac{\nabla f_i(\mathbf{x}_i^\tau)}{\|\nabla f_i(\mathbf{x}_i^\tau)\|}\right\|^2\right] \leq L^2 \rho^2 \sigma^2 (1-s)^2 \tag{22}$$

Now, we proceed to bound II:

$$II : \mathbb{E}[\|\nabla f_i(\mathbf{x}_i^\tau + \xi_i^\tau) - \nabla f_i(\mathbf{x}_i^\tau + \xi_{true}^\tau)\|]^2 \leq L^2 \mathbb{E}[\|\xi_i^\tau - \xi_{true}^\tau\|^2]$$

$$= L^2 \rho^2 \mathbb{E}\left[\left\|(1-s) \cdot \frac{\nabla f_i(\mathbf{x}_i^\tau)}{\|\nabla f_i(\mathbf{x}_i^\tau)\|} + s \cdot \left(\frac{\mathbf{d}_i^\tau}{\|\mathbf{d}_i^\tau\|} + \frac{\nabla f(\mathbf{x}_i^\tau)}{\|\nabla f(\mathbf{x}_i^\tau)\|} - \frac{\nabla f(\mathbf{x}_i^\tau)}{\|\nabla f(\mathbf{x}_i^\tau)\|}\right) - \xi_{true}^\tau\right\|^2\right]$$

$$\leq 2L^2 \rho^2 (1-s)^2 \mathbb{E}\left[\left\|\frac{\nabla f_i(\mathbf{x}_i^\tau)}{\|\nabla f_i(\mathbf{x}_i^\tau)\|} - \frac{\nabla f(\mathbf{x}_i^\tau)}{\|\nabla f(\mathbf{x}_i^\tau)\|}\right\|^2\right] + 2L^2 \rho^2 s^2 \mathbb{E}\left[\left\|\frac{\mathbf{d}_i^\tau}{\|\mathbf{d}_i^\tau\|} - \frac{\nabla f(\mathbf{x}_i^\tau)}{\|\nabla f(\mathbf{x}_i^\tau)\|}\right\|^2\right] \tag{23}$$

$$\overset{a}{\leq} 4L^2 \rho^2 (1-s)^2 + 2L^2 s^2 \rho^2 \epsilon_i^2$$

Here, (a) results from $\|\frac{\nabla f(\mathbf{x}_i^\tau)}{\|\nabla f(\mathbf{x}_i^\tau)\|}\| \leq 1$ and $\|\frac{\nabla f_i(\mathbf{x}_i^\tau)}{\|\nabla f_i(\mathbf{x}_i^\tau)\|}\| \leq 1$.

Finding an upper bound for $III$:

$$\mathbb{E}[\|\nabla f_i(\mathbf{x}_i^\tau + \xi_{true}^\tau) - \nabla f(\mathbf{x}_i^\tau + \xi_i^\tau)\|^2] =$$

$$\mathbb{E}[\|\nabla f_i(\mathbf{x}_i^\tau + \xi_{true}^\tau) - \nabla f_i(\mathbf{x}_i^\tau) + \nabla f_i(\mathbf{x}_i^\tau) - \nabla f(\mathbf{x}_i^\tau) + \nabla f(\mathbf{x}_i^\tau) - \nabla f(\mathbf{x}_i^\tau + \xi_i^\tau)\|^2]$$

$$\leq 3\mathbb{E}[\|\nabla f_i(\mathbf{x}_i^\tau + \xi_{true}^\tau) - \nabla f_i(\mathbf{x}_i^\tau)\|^2] + 3\mathbb{E}[\|\nabla f_i(\mathbf{x}_i^\tau) - \nabla f(\mathbf{x}_i^\tau)\|^2] + 3\mathbb{E}[\|\nabla f(\mathbf{x}_i^\tau) - \nabla f(\mathbf{x}_i^\tau + \xi_i^\tau)\|^2] \tag{24}$$

$$\overset{a}{\leq} 3\mathbb{E}[\|\nabla f_i(\mathbf{x}_i^\tau + \xi_{true}^\tau) - \nabla f_i(\mathbf{x}_i^\tau)\|^2] + 3\delta^2 + 3\mathbb{E}[\|\nabla f(\mathbf{x}_i^\tau) - \nabla f(\mathbf{x}_i^\tau + \xi_i^\tau)\|^2]$$

$$\overset{b}{\leq} 3L^2 \rho^2 + 3\delta^2 + 3L^2 \rho^2 = 3\delta^2 + 6L^2 \rho^2$$

(a) follows from Equation 8 in Assumption 2.
(b) follows from Assumption 1 and the perturbation being bounded by the perturbation radius $\rho$.

Putting Equation 22, 23 and 24 in Equation 21:

$$
\mathbb{E}[\|\widetilde{\mathbf{G}}_\tau\|_{\mathfrak{F}}^2] \leq 4nL^2\rho^2\sigma^2(1-s)^2 + 16nL^2\rho^2(1-s)^2 + 8L^2\rho^2 s^2 \sum_{i=1}^n \epsilon_i^2 + 12n\delta^2 + 24nL^2\rho^2
$$
$$
+ 4n\mathbb{E}[\|\frac{1}{n}\sum_{i=1}^n \nabla f_i(\mathbf{x}_i^\tau + \xi_i^\tau)\|^2]
\tag{25}
$$

Putting Equation 25 back in Equation 20:

$$
\mathbb{E}\left[\left\|\mathbf{X}^t(\mathbf{I}-\mathbf{Q})\right\|_{\mathfrak{F}}^2\right] \leq \frac{\eta^2}{(1-\sqrt{\lambda})} \sum_{\tau=0}^{t-1} \lambda^{\frac{(t-1-\tau)}{2}} (4nL^2\rho^2\sigma^2(1-s)^2 + 16nL^2\rho^2(1-s)^2 +
$$
$$
8L^2\rho^2 s^2 \sum_{i=1}^n \epsilon_i^2 + 12n\delta^2 + 24nL^2\rho^2 + 4n\mathbb{E}[\|\frac{1}{n}\sum_{i=1}^n \nabla f_i(\mathbf{x}_i^\tau + \xi_i^\tau)\|^2])
$$
$$
\leq \frac{n\eta^2}{(1-\sqrt{\lambda})^2}\left(4L^2\rho^2\sigma^2(1-s)^2 + 16nL^2\rho^2(1-s)^2 + 12\delta^2 + 24L^2\rho^2\right) + \frac{8L^2\rho^2 s^2\eta^2}{(1-\sqrt{\lambda})^2}\sum_{i=1}^n \epsilon_i^2
$$
$$
+ \frac{4n\eta^2}{(1-\sqrt{\lambda})} \sum_{\tau=0}^{t-1} \lambda^{\frac{(t-1-\tau)}{2}} \mathbb{E}[\|\frac{1}{n}\sum_{i=1}^n \nabla f_i(\mathbf{x}_i^\tau + \xi_i^\tau)\|^2])
\tag{26}
$$

Summing over $t \in \{0,\ldots,T-1\}$ with $\mathbb{E}\left[\left\|\mathbf{X}_0(\mathbf{I}-\mathbf{Q})\right\|_{\mathfrak{F}}^2\right] = 0$:

$$
\sum_{t=0}^{T-1} \mathbb{E}\left[\left\|\mathbf{X}^t(\mathbf{I}-\mathbf{Q})\right\|_{\mathfrak{F}}^2\right] \leq CT + \frac{4n\eta^2}{(1-\sqrt{\lambda})} \sum_{t=0}^{T-1}\sum_{\tau=0}^{t-1} \lambda^{\frac{(t-1-\tau)}{2}} \mathbb{E}[\|\frac{1}{n}\sum_{i=1}^n \nabla f_i(\mathbf{x}_i^\tau + \xi_i^\tau)\|^2]) \leq
$$
$$
CT + \frac{4n\eta^2}{(1-\sqrt{\lambda})} \sum_{t=0}^{T-1} \frac{1-\lambda^{(\frac{T-1-t}{2})}}{1-\sqrt{\lambda}} \mathbb{E}[\|\frac{1}{n}\sum_{i=1}^n \nabla f_i(\mathbf{x}_i^t + \xi_i^t)\|^2]) \leq
$$
$$
CT + \frac{4n\eta^2}{(1-\sqrt{\lambda})} \sum_{t=0}^{T-1} \mathbb{E}[\|\frac{1}{n}\sum_{i=1}^n \nabla f_i(\mathbf{x}_i^t + \xi_i^t)\|^2]
$$
$$
\text{where } C = \frac{n\eta^2}{(1-\sqrt{\lambda})^2}\left(4L^2\rho^2\sigma^2(1-s)^2 + 16nL^2\rho^2(1-s)^2 + 12\delta^2 + 24L^2\rho^2\right) + \frac{8L^2\rho^2 s^2\eta^2}{(1-\sqrt{\lambda})^2}\sum_{i=1}^n \epsilon_i^2
\tag{27}
$$

Dividing both sides by $n$:

$$
\sum_{t=0}^{T-1} \frac{1}{n}\mathbb{E}\left[\left\|\mathbf{X}^t(\mathbf{I}-\mathbf{Q})\right\|_{\mathfrak{F}}^2\right] \leq \frac{\eta^2 T}{(1-\sqrt{\lambda})^2}\left(4L^2\rho^2\sigma^2(1-s)^2 + 16nL^2\rho^2(1-s)^2 + 12\delta^2 + 24L^2\rho^2\right)
$$
$$
+ \frac{8L^2\rho^2 s^2\eta^2 T}{(1-\sqrt{\lambda})^2}\left(\frac{1}{n}\sum_{i=1}^n \epsilon_i^2\right) + \frac{4\eta^2}{(1-\sqrt{\lambda})} \sum_{t=0}^{T-1} \mathbb{E}[\|\frac{1}{n}\sum_{i=1}^n \nabla f_i(\mathbf{x}_i^t + \xi_i^t)\|^2]
\tag{28}
$$

This directly translates into an upper bound for the consensus error:

$$
\sum_{t=0}^{T-1} \frac{1}{n}\mathbb{E}[\sum_{i=1}^n \|\mathbf{x}_i^t - \bar{\mathbf{x}}^t\|^2] \leq \frac{4\eta^2 T}{(1-\sqrt{\lambda})^2}\left(L^2\rho^2(1-s)^2\sigma^2 + 3\delta^2 + 4L^2(1-s)^2\rho^2 +
$$
$$
2L^2 s^2\epsilon^2\rho^2 + 6L^2\rho^2\right) + \frac{4\eta^2}{(1-\sqrt{\lambda})} \sum_{t=0}^{T-1} \mathbb{E}[\|\frac{1}{n}\sum_{i=1}^n \nabla f_i(\mathbf{x}_i^t + \xi_i^t)\|^2]
\tag{29}
$$

We analyze the convergence properties of GFlat based on the following update scheme, derived from a single-step progression of the averaged model $\bar{\mathbf{x}}^t = \frac{1}{n}\sum_{i=1}^n \mathbf{x}_i^t$:

$$\bar{\mathbf{x}}^{t+1} = \bar{\mathbf{x}}^t - \eta\left(\frac{1}{n}\sum_{i=1}^n \tilde{\mathbf{g}}_i^t\right) \tag{30}$$

To begin, we reiterate Theorem 1 presented in Section 5:

**Theorem 4** *Given Assumptions 1-3, let the learning rate satisfy* $\eta \leq \frac{\sqrt{(1-\sqrt{\lambda})^2 + 16(1-\sqrt{\lambda})} - (1-\sqrt{\lambda})}{8L}$. *Then, for all* $T \geq 1$, *we have:*

$$\frac{1}{T}\sum_{t=0}^{T-1}\mathbb{E}\left[\|\nabla f(\bar{\mathbf{x}}_t)\|^2\right] \leq \frac{2}{\eta T}\left(\mathbb{E}[f(\bar{\mathbf{x}}^0)] - f^\star\right) + \sigma^2 L^2 \rho^2 (1-s)^2\left(\frac{4\eta^2 L^2}{(1-\sqrt{\lambda})^2} + \frac{\eta L}{n}\right) + \\ \delta^2 L^2\left(\frac{12\eta^2}{(1-\sqrt{\lambda})^2}\right) + \rho^2 L^2\left[\frac{8\eta^2 L^2}{(1-\sqrt{\lambda})^2}\left(2(1-s)^2 + s^2\epsilon^2 + 3\right) + 3s^2\epsilon^2 + 6s^2 + 3\right] \tag{31}$$

*Proof:* We start with the following property for a $L$-smooth function $f(\mathbf{x})$:

$$\mathbb{E}[f(\bar{\mathbf{x}}^{t+1})] \leq \mathbb{E}[f(\bar{\mathbf{x}}^t)] + \mathbb{E}[\langle \nabla f(\bar{\mathbf{x}}^t), \bar{\mathbf{x}}^{t+1} - \bar{\mathbf{x}}^t\rangle] + \frac{L}{2}\mathbb{E}[\|\bar{\mathbf{x}}^{t+1} - \bar{\mathbf{x}}^t\|^2] \tag{32}$$

$$\mathbb{E}[f(\bar{\mathbf{x}}^{t+1})] \overset{a}{\leq} \mathbb{E}[f(\bar{\mathbf{x}}^t)] - \eta\underbrace{\mathbb{E}[\langle \nabla f(\bar{\mathbf{x}}^t), \frac{1}{n}\sum_{i=1}^n \tilde{\mathbf{g}}_i^t\rangle]}_{I} + \frac{L\eta^2}{2}\mathbb{E}[\|\frac{1}{n}\sum_{i=1}^n \tilde{\mathbf{g}}_i^t\|^2] \tag{33}$$

(a) results from the update rule in Equation 30. We first start with $I$:

$$I : \mathbb{E}[\langle \nabla f(\bar{\mathbf{x}}^t), \frac{1}{n}\sum_{i=1}^n \tilde{\mathbf{g}}_i^t\rangle] = \mathbb{E}[\langle \nabla f(\bar{\mathbf{x}}^t), \frac{1}{n}\sum_{i=1}^n (\tilde{\mathbf{g}}_i^t + \nabla f_i(\mathbf{x}_i^t + \xi_i^t) - \nabla f_i(\mathbf{x}_i^t + \xi_i^t))\rangle]$$

$$= \mathbb{E}[\langle \nabla f(\bar{\mathbf{x}}^t), \frac{1}{n}\sum_{i=1}^n \nabla f_i(\mathbf{x}_i^t + \xi_i^t)\rangle] + \mathbb{E}[\langle \nabla f(\bar{\mathbf{x}}^t), \sum_{i=1}^n (\tilde{\mathbf{g}}_i^t - \nabla f_i(\mathbf{x}_i^t + \xi_i^t))\rangle]$$

$$\overset{a}{=} \mathbb{E}[\langle \nabla f(\bar{\mathbf{x}}^t), \frac{1}{n}\sum_{i=1}^n \nabla f_i(\mathbf{x}_i^t + \xi_i^t)\rangle] \overset{b}{=} \frac{1}{2}\left(\mathbb{E}\|\nabla f(\bar{\mathbf{x}}^t)\|^2 + \mathbb{E}\|\frac{1}{n}\sum_{i=1}^n \nabla f_i(\mathbf{x}_i^t + \xi_i^t)\|^2\right) - \tag{34}$$

$$\frac{1}{2}\underbrace{\mathbb{E}\left(\|\nabla f(\bar{\mathbf{x}}^t) - \frac{1}{n}\sum_{i=1}^n \nabla f_i(\mathbf{x}_t^i + \xi_i^t)\|^2\right)}_{\star}$$

(a) results from $\mathbb{E}[\tilde{\mathbf{g}}_i^t] = \nabla f_i(\mathbf{x}_i^t + \xi_i^t)$. (b) follows from $\langle \mathbf{a}, \mathbf{b}\rangle = \frac{1}{2}[\|\mathbf{a}\|^2 + \|\mathbf{b}\|^2 - \|\mathbf{a} - \mathbf{b}\|^2]$

Simplifying $\star$:

$$\|\nabla f(\bar{\mathbf{x}}^t) - \frac{1}{n}\sum_{i=1}^n \nabla f_i(\mathbf{x}_i^t + \xi_i^t)\|^2 = \|\frac{1}{n}\sum_{i=1}^n \nabla f_i(\bar{\mathbf{x}}^t) - \frac{1}{n}\sum_{i=1}^n \nabla f_i(\mathbf{x}_i^t + \xi_i^t)\|^2$$

$$\leq \frac{1}{n}\sum_{i=1}^n \|\nabla f_i(\bar{\mathbf{x}}^t) - \nabla f_i(\mathbf{x}_i^t + \xi_i^t)\|^2 \overset{a}{\leq} \frac{1}{n}\sum_{i=1}^n L^2\|\bar{\mathbf{x}}^t - \mathbf{x}_i^t - \xi_i^t\|^2 \tag{35}$$

$$\overset{b}{\leq} \frac{L^2}{n}\sum_{i=1}^n (\|\bar{\mathbf{x}}^t - \mathbf{x}_i^t\|^2 + \|\xi_i^t\|^2)$$

(a) follows from Assumption 1. (b) results from the inequity $\|a - b\|^2 \leq \|a\|^2 + \|b\|^2$.

Now, we define an upper bound for $\|\xi_i^t\|^2$:

$$
\begin{aligned}
\|\xi_i^t\|^2 &= \rho^2 \|s. \frac{\mathbf{d}_i^t}{\|\mathbf{d}_i^t\|} + (1-s). \frac{\mathbf{g}_i^t}{\|\mathbf{g}_i^t\|}\|^2 = \rho^2 \|s \left( \frac{\mathbf{d}_i^t}{\|\mathbf{d}_i^t\|} - \frac{\nabla f(\mathbf{x}_i^t)}{\|\nabla f(\mathbf{x}_i^t)\|} \right) + s \frac{\nabla f(\mathbf{x}_i^t)}{\|\nabla f(\mathbf{x}_i^t)\|} + (1-s). \frac{\mathbf{g}_i^t}{\|\mathbf{g}_i^t\|}\|^2 \\
&\leq \rho^2 (3s^2\epsilon^2 + 3s^2 + 3(1-s)^2) \leq 3\rho^2 s^2 \epsilon_i^2 + 6\rho^2 s^2 + 3\rho^2
\end{aligned}
\tag{36}
$$

Putting this result in Equation 35:

$$
\begin{aligned}
\|\nabla f(\bar{\mathbf{x}}^t) - \frac{1}{n} \sum_{i=1}^n \nabla f_i(\mathbf{x}_i^t + \xi_i^t)\|^2 &\leq \frac{L^2}{n} \sum_{i=1}^n (\|\bar{\mathbf{x}}^t - \mathbf{x}_i^t\|^2 + \|\xi_i^t\|^2) \leq \frac{L^2}{n} \sum_{i=1}^n (\|\bar{\mathbf{x}}^t - \mathbf{x}_i^t\|^2) \\
&+ 3L^2\rho^2 s^2 \epsilon^2 + 6L^2\rho^2 s^2 + 3L^2\rho^2
\end{aligned}
\tag{37}
$$

Note that $\frac{1}{n} \sum_{i=1}^n \epsilon_i^2 \leq \epsilon^2$, where $\epsilon = \max_{1 \leq i \leq n} \epsilon_i$.

Substituting Equation 37 into Equation 34:

$$
\begin{aligned}
\mathbb{E}[\langle \nabla f(\bar{\mathbf{x}}^t), \frac{1}{n} \sum_{i=1}^n \widetilde{\mathbf{g}}_i^t \rangle] &\geq \frac{1}{2} \mathbb{E}[\|\nabla f(\bar{\mathbf{x}}^t)\|^2 + \|\frac{1}{n} \sum_{i=1}^n \nabla f_i(\mathbf{x}_i^t + \xi_i^t)\|^2] - \frac{L^2}{2n} \mathbb{E}[\sum_{i=1}^n (\|\bar{\mathbf{x}}^t - \mathbf{x}_i^t\|^2)] - \\
\frac{3L^2\rho^2 s^2 \epsilon^2}{2} &- 3L^2\rho^2 s^2 - \frac{3L^2\rho^2}{2}
\end{aligned}
\tag{38}
$$

Substituting Equation 38 into Equation 33:

$$
\begin{aligned}
\mathbb{E}[f(\bar{\mathbf{x}}^{t+1})] &\leq \mathbb{E}[f(\bar{\mathbf{x}}^t)] - \frac{\eta}{2} \mathbb{E}[\|\nabla f(\bar{\mathbf{x}}^t)\|^2] - \frac{\eta}{2} \mathbb{E}[\|\frac{1}{n} \sum_{i=1}^n \nabla f_i(\mathbf{x}_i^t + \xi_i^t)\|^2] \\
&+ \frac{\eta L^2}{2} \mathbb{E}[\frac{1}{n} \sum_{i=1}^n (\|\bar{\mathbf{x}}^t - \mathbf{x}_i^t\|^2)] + 3\eta L^2 \rho^2 \left( \frac{s^2\epsilon^2}{2} + s^2 + \frac{1}{2} \right) + \frac{L\eta^2}{2} \mathbb{E}[\|\frac{1}{n} \sum_{i=1}^n \widetilde{\mathbf{g}}_i^t\|^2]
\end{aligned}
\tag{39}
$$

Rearranging the terms and dividing by $\frac{\eta}{2} > 0$ to find the bound for $\mathbb{E}[\|\nabla f(\bar{\mathbf{x}}^t)\|^2]$:

$$
\begin{aligned}
\mathbb{E}[\|\nabla f(\bar{\mathbf{x}}^t)\|^2] &\leq \frac{2}{\eta} \left( \mathbb{E}[f(\bar{\mathbf{x}}^t)] - \mathbb{E}[f(\bar{\mathbf{x}}^{t+1})] \right) - \mathbb{E}[\|\frac{1}{n} \sum_{i=1}^n \nabla f_i(\mathbf{x}_i^t + \xi_i^t)\|^2] \\
&+ L^2 \mathbb{E}[\frac{1}{n} \sum_{i=1}^n (\|\bar{\mathbf{x}}^t - \mathbf{x}_i^t\|^2)] + 6L^2\rho^2 \left( \frac{s^2\epsilon^2}{2} + s^2 + \frac{1}{2} \right) + \eta L \mathbb{E}[\|\frac{1}{n} \sum_{i=1}^n \widetilde{\mathbf{g}}_i^t\|^2]
\end{aligned}
\tag{40}
$$

We first bound $\mathbb{E}[\|\frac{1}{n} \sum_{i=1}^n \widetilde{\mathbf{g}}_i^t\|^2]$:

$$
\begin{aligned}
\mathbb{E}[\|\frac{1}{n} \sum_{i=1}^n \widetilde{\mathbf{g}}_i^t\|^2] &= \mathbb{E}[\|\frac{1}{n} \sum_{i=1}^n (\widetilde{\mathbf{g}}_i^t - \nabla f_i(\mathbf{x}_i^\tau + \xi_i^\tau))\|^2] + \mathbb{E}[\|\frac{1}{n} \sum_{i=1}^n (\nabla f_i(\mathbf{x}_i^\tau + \xi_i^\tau))\|^2] \\
&\overset{a}{=} \frac{1}{n^2} \sum_{i=1}^n \mathbb{E}[\|\widetilde{\mathbf{g}}_i^t - \nabla f_i(\mathbf{x}_i^\tau + \xi_i^\tau)\|^2] + \mathbb{E}[\|\frac{1}{n} \sum_{i=1}^n (\nabla f_i(\mathbf{x}_i^\tau + \xi_i^\tau))\|^2] \overset{b}{\leq} \frac{L^2\rho^2\sigma^2(1-s)^2}{n} + \\
&\mathbb{E}[\|\frac{1}{n} \sum_{i=1}^n (\nabla f_i(\mathbf{x}_i^\tau + \xi_i^\tau))\|^2]
\end{aligned}
\tag{41}
$$

(a) holds because $\widetilde{\mathbf{g}}_i^t - \nabla f_i(\mathbf{x}_i^\tau + \xi_i^\tau)$ are independent vectors with mean 0 Yu et al. (2019), and (b) follows from equation 22.

Using the above result in Equation 40, we have:

$$
\begin{aligned}
\mathbb{E}[\|\nabla f(\bar{\mathbf{x}}^t)\|^2] \leq{}& \frac{2}{\eta}\left(\mathbb{E}[f(\bar{\mathbf{x}}^t)] - \mathbb{E}[f(\bar{\mathbf{x}}^{t+1})]\right) - \mathbb{E}[\|\frac{1}{n}\sum_{i=1}^n \nabla f_i(\mathbf{x}_i^t + \xi_i^t)\|^2] + L^2\mathbb{E}[\frac{1}{n}\sum_{i=1}^n \|\bar{\mathbf{x}}^t - \mathbf{x}_i^t\|^2] \\
&+ 6L^2\rho^2\left(\frac{s^2\epsilon^2}{2} + s^2 + \frac{1}{2}\right) + \eta L\left(\frac{L^2\rho^2\sigma^2(1-s)^2}{n} + \mathbb{E}[\|\frac{1}{n}\sum_{i=1}^n \nabla f_i(\mathbf{x}_i^t + \xi_i^t)\|^2]\right)
\end{aligned}
\tag{42}
$$

Summing over $t \in 0, 1..., T-1$ and dividing by $T$, we have:

$$
\begin{aligned}
\frac{1}{T}\sum_{t=0}^{T-1}\mathbb{E}[\|\nabla f(\bar{\mathbf{x}}^t)\|^2] \leq{}& \frac{2}{\eta T}\left(f(\bar{\mathbf{x}}^0) - f(\bar{\mathbf{x}}^\star)\right) - \frac{1}{T}\sum_{t=0}^{T-1}\left(\mathbb{E}[\|\frac{1}{n}\sum_{i=1}^n \nabla f_i(\mathbf{x}_i^t + \xi_i^t)\|^2]\right) \\
&+ L^2\left(\frac{1}{T}\sum_{t=0}^{T-1}\mathbb{E}[\frac{1}{n}\sum_{i=1}^n(\|\bar{\mathbf{x}}^t - \mathbf{x}_i^t\|^2)]\right) + 6L^2\rho^2\left(\frac{s^2\epsilon^2}{2} + s^2 + \frac{1}{2}\right) + \\
&\eta L\left(\frac{L^2\rho^2\sigma^2(1-s)^2}{n} + \frac{1}{T}\sum_{t=0}^{T-1}\mathbb{E}[\|\frac{1}{n}\sum_{i=1}^n \nabla f_i(\mathbf{x}_i^t + \xi_i^t)\|^2]\right)
\end{aligned}
\tag{43}
$$

Using the result of Lemma 3 in the above Equation and rearranging the terms:

$$
\begin{aligned}
\frac{1}{T}\sum_{t=0}^{T-1}\mathbb{E}[\|\nabla f(\bar{\mathbf{x}}^t)\|^2] \leq{}& \frac{2}{\eta T}\left(f(\bar{\mathbf{x}}^0) - f(\bar{\mathbf{x}}^\star)\right) + \left(\eta L - 1 + \frac{4\eta^2 L^2}{1-\sqrt{\lambda}}\right) \\
&\left(\frac{1}{T}\sum_{t=0}^{T-1}(\mathbb{E}[\|\frac{1}{n}\sum_{i=1}^n \nabla f_i(\mathbf{x}_i^t + \xi_i^t)\|^2])\right) + \frac{4\eta^2 L^2}{(1-\sqrt{\lambda})^2}\left(L^2\rho^2(1-s)^2\sigma^2 + 3\delta^2 + 4L^2(1-s)^2\rho^2\right. \\
&\left. + 2L^2 s^2\epsilon^2\rho^2 + 6L^2\rho^2\right) + \eta L\left(\frac{L^2\rho^2\sigma^2(1-s)^2}{n}\right) + 6L^2\rho^2\left(\frac{s^2\epsilon^2}{2} + s^2 + \frac{1}{2}\right) \\
={}& \frac{2}{\eta T}\left(f(\bar{\mathbf{x}}^0) - f(\bar{\mathbf{x}}^\star)\right) + \left(\eta L - 1 + \frac{4\eta^2 L^2}{1-\sqrt{\lambda}}\right)\left(\frac{1}{T}\sum_{t=0}^{T-1}(\mathbb{E}[\|\frac{1}{n}\sum_{i=1}^n \nabla f_i(\mathbf{x}_i^t + \xi_i^t)\|^2])\right) + \\
&L^2\rho^2(1-s)^2\sigma^2\left(\frac{4\eta^2 L^2}{(1-\sqrt{\lambda})^2} + \frac{\eta L}{n}\right) + \left(\frac{12\eta^2 L^2}{(1-\sqrt{\lambda})^2}\right)\delta^2 \\
&+ L^2\rho^2\left[\frac{8\eta^2 L^2}{(1-\sqrt{\lambda})^2}\left(2(1-s)^2 + s^2\epsilon^2 + 3\right) + 3s^2\epsilon^2 + 6s^2 + 3\right]
\end{aligned}
\tag{44}
$$

When $\eta L - 1 + \frac{4\eta^2 L^2}{1-\sqrt{\lambda}} \leq 0$:

$$
\begin{aligned}
\frac{1}{T}\sum_{t=0}^{T-1}\mathbb{E}[\|\nabla f(\bar{\mathbf{x}}^t)\|^2] \leq{}& \frac{2}{\eta T}\left(f(\bar{\mathbf{x}}^0) - f^\star\right) + \sigma^2 L^2\rho^2(1-s)^2\left(\frac{4\eta^2 L^2}{(1-\sqrt{\lambda})^2} + \frac{\eta L}{n}\right) + \\
&\delta^2 L^2\left(\frac{12\eta^2}{(1-\sqrt{\lambda})^2}\right) + \rho^2 L^2\left[\frac{8\eta^2 L^2}{(1-\sqrt{\lambda})^2}\left(2(1-s)^2 + s^2\epsilon^2 + 3\right) + 3s^2\epsilon^2 + 6s^2 + 3\right]
\end{aligned}
\tag{45}
$$

## B.3 Discussion on the Step Size

The condition for Equation 45 to be true is $\eta L - 1 + \frac{4\eta^2 L^2}{1-\sqrt{\lambda}} \leq 0$. Upon solving this inequality with the fact that $\eta > 0$, we have:

$$
\eta \leq \frac{\sqrt{(1-\sqrt{\lambda})^2 + 16(1-\sqrt{\lambda})} - (1-\sqrt{\lambda})}{8L}
\tag{46}
$$

### B.4 Proof for Corollary 2

According to Equation (45), on the right hand side, there are four terms with different coefficients with respect to the step size $\eta = \mathcal{O}\left(\sqrt{\frac{n}{T}}\right)$ and perturbation radius $\rho = \mathcal{O}\left(\sqrt{\frac{1}{T}}\right)$. We separately investigate each term:

$$\frac{2}{\eta T}\left(f(\bar{\mathbf{x}}^0) - f^\star\right) = \mathcal{O}\left(\frac{1}{\sqrt{nT}}\right) \tag{47}$$

$$L^2\rho^2(1-s)^2\left(\frac{4\eta^2 L^2}{(1-\sqrt{\lambda})^2} + \frac{\eta L}{n}\right)\sigma^2 = \mathcal{O}\left(\frac{n}{T^2} + \frac{1}{n^{1/2}T^{3/2}}\right) \tag{48}$$

$$\left(\frac{12\eta^2 L^2}{(1-\sqrt{\lambda})^2}\right)\delta^2 = \mathcal{O}\left(\frac{n}{T}\right) \tag{49}$$

$$L^2\left[\frac{8\eta^2 L^2}{(1-\sqrt{\lambda})^2}\left(2(1-s)^2 + s^2\epsilon^2 + 3\right) + 3s^2\epsilon^2 + 6s^2 + 3\right]\rho^2 = \mathcal{O}\left(\frac{n}{T^2} + \frac{1}{T}\right) \tag{50}$$

By omitting $n$ in higher-order terms (since $T >> n$) and combining the results of all the above equations, the convergence rate is as follows:

$$\frac{1}{T}\sum_{t=0}^{T-1}\mathbb{E}\left[\left\|\nabla f\left(\bar{\mathbf{x}}^t\right)\right\|^2\right] \leq \mathcal{O}\left(\frac{f(\bar{\mathbf{x}}^0) - f^\star}{\sqrt{nT}} + \frac{(1-s)\sigma^2}{T^{3/2}} + \frac{\delta^2}{T} + \frac{\epsilon^2}{T}\right) \tag{51}$$

This implies that when $T$ is sufficiently large, GFlat results in a convergence rate of $\mathcal{O}\left(\frac{1}{\sqrt{nT}}\right)$.

## C Algorithmic Details

The authors in Lin et al. (2021) propose Quasi-Global Momentum (QGM) buffer, which improves the local momentum acceleration in the presence of data heterogeneity. Specifically, QGM updates the momentum buffer by calculating the difference between two consecutive models, $\mathbf{x}_i^{t+1}$ and $\mathbf{x}_i^t$, to locally approximate the global optimization direction. The update rule for QGM is given by the following equation:

$$\text{QGM:}\ \mathbf{x}_i^{t+1} = \sum_{j\in\mathcal{N}(i)} w_{ij}[\mathbf{x}_j^t - \eta(\mathbf{g}_j^t + \beta\mathbf{m}_j^{t-1})];\ \ \mathbf{m}_i^t = \mu\mathbf{m}_i^{t-1} + (1-\mu)\frac{\mathbf{x}_i^t - \mathbf{x}_i^{(t+1)}}{\eta}. \tag{52}$$

To further enhance performance, we introduce a variant of our approach called Q-GFlat. For implementation details, please refer to Algorithm 2.

---

**Algorithm 2** Q-GFlat

---

**Input:** Each client $i \in [1,n]$ initializes model parameters $\mathbf{x}_i$, step size $\eta$, perturbation radius $\rho$, scaling factor $s \in [0,1]$, mixing matrix $\mathbf{W} = [w_{ij}]_{i,j \in [1,n]}$, $\mathcal{N}(i)$ represents neighbors of $i$, momentum coefficients $\beta, \mu$.

**procedure** TRAIN( ) for $\forall i$
1.    **for** t $= 1, 2, \ldots, T$ **do**
2.       $\mathbf{g}_i^t = \nabla F_i(\mathbf{x}_i^t; \mathcal{B}_i^t)$ for $\mathcal{B}_i^t \sim D_i$
3.       $\mathbf{m}_i^t = \beta \hat{\mathbf{m}}_i^{t-1} + \mathbf{g}_i^t$
4.       $\mathbf{d}_i^t = \mathbf{x}_i^{t-1} - \mathbf{x}_i^t$
5.       $\xi_i^t = \rho\big(s.\frac{\mathbf{d}_i^t}{\|\mathbf{d}_i^t\|} + (1-s).\frac{\mathbf{m}_i^t}{\|\mathbf{m}_i^t\|}\big)$
6.       $\widetilde{\mathbf{g}}_i^t = \nabla F_i(\mathbf{x}_i^t + \xi_i^t; \mathcal{B}_i^t)$
7.       $\mathbf{x}_i^{(t+1/2)} = \mathbf{x}_i^{(t)} - \eta \widetilde{\mathbf{g}}_i^t$
8.       SENDRECEIVE$(\mathbf{x}_i^{(t+1/2)})$
9.       $\mathbf{x}_i^{t+1} = \sum_{j \in \mathcal{N}_i^{(t)}} w_{ij} \mathbf{x}_j^{(t+1/2)}$
10.      $\hat{\mathbf{m}}_i^t = \mu \hat{\mathbf{m}}_i^{t-1} + (1-\mu)\mathbf{d}_i^t$
11.   **end**
**return** $\mathbf{x}_i^T$

---

# D   Setup Details

## D.1   Datasets and Models

For ImageNet Deng et al. (2009), the terms of access and details related to the license are available at `https://image-net.org/download.php`. We download Imagenette Husain (2018) from `https://github.com/fastai/imagenette`, which has Apache License 2.0.

All models use Evonorm Liu et al. (2020); Hsieh et al. (2020a) as a normalization layer, as it is shown to be better suited for decentralized learning with non-IID data. We use the standard ResNet-18 and ResNet-20 architectures He et al. (2016) with 11M and 0.27M trainable model parameters, respectively. For MobileNet-V2, we use the architecture with 2.3M parameters Sandler et al. (2018).

## D.2   Non-IID Distribution

Similar to prior works Choudhary et al. (2025); Aketi et al. (2023b;a); Lin et al. (2021), we use the Dirichlet distribution to simulate different levels of data heterogeneity in our setup Hsu et al. (2019). Figure 6 shows the number of data points from each class of CIFAR-10 that are allocated to different clients for $\alpha = 0.01, 0.001$. While $\alpha = 0.01$ allows most clients to have samples from two different classes, $\alpha = 0.001$ is the most extreme form of non-IIDness, with clients having samples from only one class.

## D.3   Graph Topologies

We perform our experiments for two different graph topologies, ring and torus. Ring is one of the most sparsely connected topology and has 2 peers/neighbors per client, while torus has 4. Please refer to Figure 7 for a visualization of these.

Table 6: Learning rate ($\eta$), batch size per client, and the number of epochs for all the experiments.

| Dataset | CIFAR-10 | CIFAR-100 | Imagenette | ImageNet |
|---|---|---|---|---|
| Learning Rate ($\eta$) | 0.1 | 0.1 | 0.01 | 0.01 |
| Epochs | 200 | 100 | 100 | 50 |
| Batch-Size/Client | 32 | 20 | 32 | 64 |

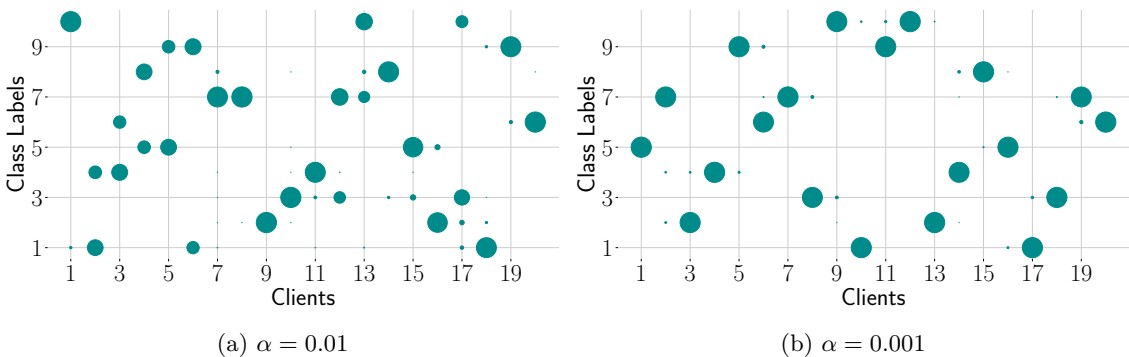

(a) $\alpha = 0.01$          (b) $\alpha = 0.001$

Figure 6: Test accuracy for the CIFAR-10 trained on ResNet-20 architecture distributed with various levels of heterogeneity ($\alpha$) over 20 clients.

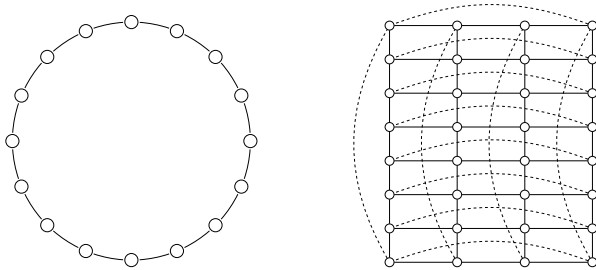

Figure 7: Ring Graph (left), and Torus Graph (right).

## D.4  Hyperparameters

The initial model and all training hyperparameters are synchronized at the start of training, and communication between clients is assumed to be synchronous. The initial learning rate is set according to Table 6, with a decay factor of 10 applied at 50% and 75% of training. We use a weight decay of 0.0001, and training proceeds for a fixed number of epochs as specified in Table 6. Quasi-Global Momentum (QGM) uses the Nesterov momentum with a coefficient of 0.9. For SADDLe and Q-SADDLe Choudhary et al. (2025), we adopt the same perturbation radius $\rho$ as reported in the original paper. Additionally, we perform a grid search over $\{0.05, 0.1, 0.2\}$ for $\rho$ and $\{0.2, 0.5, 0.7, 1\}$ for the scaling factor $s$, selecting the best values based on performance (see Table 7 for details). As noted in Remark 1 in Section 5, $0.5 <= s <= 1$ yields the best results. Interestingly, when $s = 1$, GFlat becomes more computationally efficient than SADDLe, as it eliminates the ascent step required to compute the local perturbation. We report the test accuracy of the consensus model averaged over three randomly selected seeds. All experiments are conducted using NVIDIA A40 GPUs.

Table 7: Perturbation Radius ($\rho$) and Scaling Factor ($s$) for GFlat and Q-GFlat.

| Dataset | CIFAR-10 | CIFAR-100 | Imagenette | ImageNet |
|---|---|---|---|---|
| Perturbation Radius ($\rho$) | 0.1 | 0.2 | 0.05 | 0.05 |
| *GFlat*: Scaling Factor ($s$) | 0.5 | 0.7 | 0.5, 0.7 | 1 |
| *Q-GFlat*: Scaling Factor ($s$) | 0.7, 1 | 1 | 0.7, 1 | - |

## E  Additional Discussion on Local and Global Flatness

In decentralized learning, each client's objective $f_i(\mathbf{x})$ captures only a partial view of the global objective $f(\mathbf{x}) = \frac{1}{n} f_i(\mathbf{x})$. Under IID data, these objectives are well aligned: as shown in Figure 8, the eigenvalue spectra

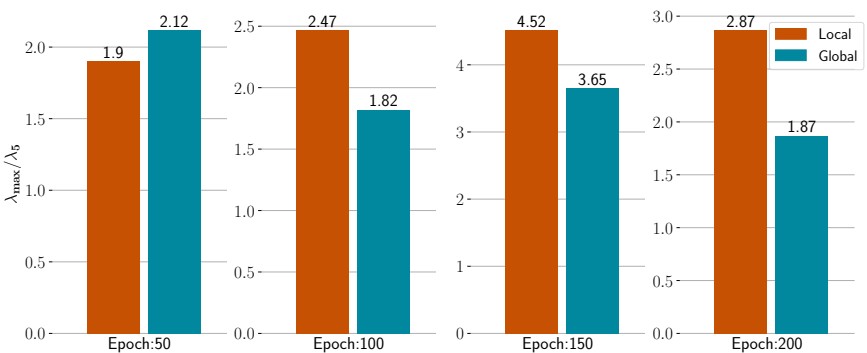

Figure 8: Ratio of largest to 5th largest eigenvalue ($\lambda_{max}/\lambda_5$) of the Hessian of the local and globally averaged models at 4 different training stages, under an IID data distribution across 10 clients on the CIFAR-10 dataset.

of local and global Hessians are nearly identical, indicating consistent curvature across clients. However, under non-IID data, the curvature of these local landscapes can differ dramatically, i.e., regions that appear flat for one client may correspond to sharp directions in the global loss. Consequently, optimizing for local flatness alone does not guarantee global flatness, as done in prior methods such as SADDLe Choudhary et al. (2025). In fact, aggregating updates from locally flat but mutually misaligned regions may yield globally sharp solutions that hinder consensus and generalization. This phenomenon is clearly evident in Figure 3, where the discrepancy between local and global flatness widens markedly under heterogeneous data partitions.

We attribute this misalignment to the fact that each client estimates its worst-case perturbation using only its own local gradient $\mathbf{g}_i$, which captures limited curvature information. Ideally, one would instead compute perturbations based on the global gradient ($\bar{\mathbf{g}}^t = 1/n \sum_{i=1}^{n} \mathbf{g}_i^t$), ensuring that every client's update direction is consistent with the global curvature. While this would align local SAM steps with global flatness, such synchronization fundamentally violates the decentralized setup, requiring full communication among all clients at every iteration.

## E.1  Theoretical Analysis

To overcome the above mentioned limitation, we introduce a locally approximated global direction $\mathbf{d}_i^t$, which allows each client to estimate $\bar{\mathbf{g}}^t$ using model differences across consecutive training iterations. This approximation enables clients to incorporate a proxy of the global update into their local SAM objectives, effectively aligning local and global flatness without incurring any communication overhead. We refer to the discrepancy between the approximated and true global directions, $\|\mathbf{d}_i^t - \eta\bar{\mathbf{g}}^t\|$, as the perturbation deviation, which quantifies how closely each client's local displacement tracks the global gradient direction. Lemma 5 formalizes this intuition, showing that the perturbation deviation decays asymptotically at rate $\mathcal{O}(1/T)$.

**Lemma 5** *Suppose Assumptions 1–3 hold. Let the stepsize and perturbation radius satisfy $\eta = \mathcal{O}\sqrt{\frac{n}{T}}$ and $\rho = \mathcal{O}(\sqrt{T})$. Then*

$$\frac{1}{T}\sum_{t=1}^{T}\frac{1}{n}\sum_{i=1}^{n}\left\|\mathbf{d}_i^t - \eta\bar{\mathbf{g}}^t\right\|^2 \;\leq\; \mathcal{O}\left(\frac{1}{T}\right). \tag{53}$$

*Proof*: Recall the parameter update rule and our approximated global direction $\mathbf{d}_i^t$ from Algorithm 1:

$$\mathbf{x}_i^t = \sum_{j\in\mathcal{N}_i^{(t)}} w_{ij}(\mathbf{x}_j^{t-1} - \eta\tilde{\mathbf{g}}_j^{t-1}) \tag{54}$$

$$\mathbf{d}_i^t = \mathbf{x}_i^{t-1} - \mathbf{x}_i^t \;=\; \eta\sum_j w_{ij}\,\tilde{\mathbf{g}}_j^{t-1} \;+\; \left(\mathbf{x}_i^{t-1} - \sum_j w_{ij}\mathbf{x}_j^{t-1}\right). \tag{55}$$

Add and subtract $\eta \sum_j w_{ij} \mathbf{g}_j^t$ and $\eta \sum_j w_{ij} \mathbf{g}_j^{t-1}$:

$$\mathbf{d}_i^t - \eta \bar{\mathbf{g}}^t = \underbrace{\eta \sum_j w_{ij} \big(\mathbf{g}_j^{t-1} - \mathbf{g}_j^t\big)}_{A} + \underbrace{\eta \Big(\sum_j w_{ij} \mathbf{g}_j^t - \tfrac{1}{n} \sum_j \mathbf{g}_j^t\Big)}_{B} + \underbrace{\Big(\mathbf{x}_i^{t-1} - \sum_j w_{ij} \mathbf{x}_j^{t-1}\Big)}_{C}.$$
$$+ \underbrace{\eta \sum_j w_{ij} \big(\widetilde{\mathbf{g}}_j^{t-1} - \mathbf{g}_j^{t-1}\big)}_{D} \tag{56}$$

$$\left\| \mathbf{d}_i^t - \eta \bar{\mathbf{g}}^t \right\|^2 \le 4\big(\|A\|^2 + \|B\|^2 + \|C\|^2 + \|D\|^2\big). \tag{57}$$

We start by bounding A. Through Assumption 1, $\|\mathbf{g}_j^{t-1} - \mathbf{g}_j^t\| \le L\|\mathbf{x}_j^{t-1} - \mathbf{x}_j^t\| = L\|\mathbf{d}_j^t\|$. Hence, using Jensen inequality,

$$\|A\|^2 = \eta^2 \Big\| \sum_j w_{ij}(\mathbf{g}_j^{t-1} - \mathbf{g}_j^t) \Big\|^2 \le \eta^2 L^2 \sum_j w_{ij}\|\mathbf{d}_j^t\|^2 \le \eta^2 L^2 \sum_j \|\mathbf{d}_j^t\|^2. \tag{58}$$

Based on notations in Equation 15, $Q = \frac{1}{n}\mathbf{1}\mathbf{1}^\top$ and $\mathbf{G}^t \triangleq [\mathbf{g}_1^t, \mathbf{g}_2^t, ..., \mathbf{g}_n^t]$. We will also use the stacked vector notation only to apply $\|\mathbf{W} - Q\|_2$; non-neighbors simply have $w_{ij} = 0$. With standard basis vector $\mathbf{e}_i^\top$ and $\mathbf{W} - Q\|_2 \le \sqrt{\lambda}$, we have:

$$\|B\| = \eta \, \mathbf{e}_i^\top (\mathbf{W} - Q) \, \mathbf{G}^t \le \eta \|\mathbf{W} - Q\|_2 \|\mathbf{G}^t\|_2 \le \eta \sqrt{\lambda} \Big( \sum_{j=1}^n \|\mathbf{g}_j^t\|^2 \Big)^{1/2}. \tag{59}$$

For each $j$, add and subtract $\nabla f_j(\bar{\mathbf{x}}^t)$ and use the inequality $\|u + v\|^2 \le 2\|u\|^2 + 2\|v\|^2$:

$$\begin{aligned}
\|\mathbf{g}_j^t\|^2 &= \left\| \nabla f_j(\bar{\mathbf{x}}^t) + \big(\nabla f_j(\mathbf{x}_j^t) - \nabla f_j(\bar{\mathbf{x}}^t)\big) \right\|^2 \\
&\le 2 \left\| \nabla f_j(\bar{\mathbf{x}}^t) \right\|^2 + 2 \left\| \nabla f_j(\mathbf{x}_j^t) - \nabla f_j(\bar{\mathbf{x}}^t) \right\|^2.
\end{aligned} \tag{60}$$

Summing equation 60 over $j = 1, \dots, n$ yields

$$\sum_{j=1}^n \|\nabla f_j(\mathbf{x}_j^t)\|^2 \le 2 \sum_{j=1}^n \left\| \nabla f_j(\bar{\mathbf{x}}^t) \right\|^2 + 2 \sum_{j=1}^n \left\| \nabla f_j(\mathbf{x}_j^t) - \nabla f_j(\bar{\mathbf{x}}^t) \right\|^2. \tag{61}$$

By L-smoothness (Assumption-1), we have:

$$\sum_{j=1}^n \left\| \nabla f_j(\mathbf{x}_j^t) - \nabla f_j(\bar{\mathbf{x}}^t) \right\|^2 \le L^2 \sum_{j=1}^n \|\mathbf{x}_j^t - \bar{\mathbf{x}}^t\|^2 = L^2 \Delta_t \tag{62}$$

$$\sum_{j=1}^n \left\| \nabla f_j(\bar{\mathbf{x}}^t) \right\|^2 = \sum_{j=1}^n \left\| \nabla f_j(\bar{\mathbf{x}}^t) - \nabla f(\bar{\mathbf{x}}^t) + \nabla f(\bar{\mathbf{x}}^t) \right\|^2 \le 2n\delta^2 + 2n\|\nabla f(\bar{\mathbf{x}}^t)\|^2 \tag{63}$$

$$\sum_{j=1}^n \|\mathbf{g}_j^t\|^2 = \sum_{j=1}^n \|\nabla f_j(\mathbf{x}_j^t)\|^2 \le 4n\Big(\|\nabla f(\bar{\mathbf{x}}^t)\|^2 + \delta^2\Big) + 2L^2 \Delta_t \tag{64}$$

Substituting the above in equation 59:

$$\|B\|^2 \le \eta^2 \lambda \Big[ 4n\big(\|\nabla f(\bar{\mathbf{x}}^t)\|^2 + \delta^2\big) + 2L^2 \Delta_t \Big]. \tag{65}$$

Now we bound C, where $C = \mathbf{x}_i^{t-1} - \sum_j w_{ij}\mathbf{x}_j^{t-1}$. We stack all client models into $\mathbf{X}^{t-1} = [(\mathbf{x}_1^{t-1})^\top; \ldots; (\mathbf{x}_n^{t-1})^\top] \in \mathbb{R}^{n \times d}$, so that the $i$-th row corresponds to client $i$. Then $C$ can be written as

$$C = \mathbf{e}_i^\top (\mathbf{I} - \mathbf{W})\mathbf{X}^{t-1}. \tag{66}$$

Here, $\mathbf{e}_i^\top \in \mathbb{R}^n$ is the standard basis vector. Since $\mathbf{W}$ is doubly stochastic, we have $\mathbf{WQ} = \mathbf{QW} = \mathbf{Q}$, and therefore

$$(\mathbf{I} - \mathbf{W}) = (\mathbf{I} - \mathbf{W})(\mathbf{I} - \mathbf{Q}) \quad \Longleftrightarrow \quad (\mathbf{W} - \mathbf{Q}) = \mathbf{W}(\mathbf{I} - \mathbf{Q}).$$

$$C = \mathbf{e}_i^\top (\mathbf{I} - \mathbf{W})\mathbf{X}^{t-1} = \mathbf{e}_i^\top (\mathbf{I} - \mathbf{W})(\mathbf{I} - \mathbf{Q})\mathbf{X}^{t-1} = \mathbf{e}_i^\top (\mathbf{W} - \mathbf{Q})\big(\mathbf{X}^{t-1} - \bar{\mathbf{x}}^{t-1}\mathbf{1}^\top\big) \tag{67}$$

By Assumption 3, $\|\mathbf{W} - \mathbf{Q}\|_2 \le \sqrt{\lambda}$:

$$\|C\| = \|\mathbf{e}_i^\top (\mathbf{W} - \mathbf{Q})(\mathbf{X}^{t-1} - \bar{\mathbf{x}}^{t-1}\mathbf{1}^\top)\| \le \sqrt{\lambda}\,\|\mathbf{X}^{t-1} - \bar{\mathbf{x}}^{t-1}\mathbf{1}^\top\|_F.$$

Since $\|\mathbf{A}\|_F^2 = \sum_j \|\mathbf{a}_j\|^2$ for the rows $\mathbf{a}_j$ of $\mathbf{A}$, this yields

$$\|C\|^2 \le \lambda \sum_{j=1}^n \|\mathbf{x}_j^{t-1} - \bar{\mathbf{x}}^{t-1}\|^2 = \lambda\,\Delta_{t-1}. \tag{68}$$

Now we bound D with L-smoothness and bounded perturbation:

$$\left\| \eta \sum_j w_{ij}\big(\widetilde{\mathbf{g}}_j^{t-1} - \mathbf{g}_j^{t-1}\big) \right\|^2 \le \eta^2 L^2 \rho^2 \tag{69}$$

Now we bound $\sum_j \|\mathbf{d}_j^t\|^2$. From equation 55, we add and subtract the unperturbed gradients to obtain:

$$\begin{aligned}
\mathbf{d}_j^t &= \eta \sum_k w_{jk}\widetilde{\mathbf{g}}_k^{t-1} + \left(\mathbf{x}_j^{t-1} - \sum_k w_{jk}\mathbf{x}_k^{t-1}\right) \\
&= \eta \sum_k w_{jk}\mathbf{g}_k^t + \eta \sum_k w_{jk}\big(\mathbf{g}_k^{t-1} - \mathbf{g}_k^t\big) + \eta \sum_k w_{jk}\big(\widetilde{\mathbf{g}}_k^{t-1} - \mathbf{g}_k^{t-1}\big) + \left(\mathbf{x}_j^{t-1} - \sum_k w_{jk}\mathbf{x}_k^{t-1}\right).
\end{aligned} \tag{70}$$

With $A_j := \eta \sum_k w_{jk}\mathbf{g}_k^t$, $B_j := \eta \sum_k w_{jk}\big(\mathbf{g}_k^{t-1} - \mathbf{g}_k^t\big)$, $C_j := \mathbf{x}_j^{t-1} - \sum_k w_{jk}\mathbf{x}_k^{t-1}$, $D_j := \eta \sum_k w_{jk}\big(\widetilde{\mathbf{g}}_k^{t-1} - \mathbf{g}_k^{t-1}\big)$ we have,

$$\sum_{j=1}^n \|\mathbf{d}_j^t\|^2 \le 4\sum_j \|A_j\|^2 + 4\sum_j \|B_j\|^2 + 4\sum_j \|C_j\|^2 + 4\sum_j \|D_j\|^2 \tag{71}$$

$$\sum_j \|A_j\|^2 = \eta^2 \|\mathbf{W}\mathbf{G}^t\|_F^2 \le \eta^2 \|\mathbf{W}\|_2^2 \|\mathbf{G}^t\|_F^2 \le \eta^2 \sum_{k=1}^n \|\mathbf{g}_k^t\|^2, \tag{72}$$

since $\|\mathbf{W}\|_2 \le 1$ for a doubly stochastic $\mathbf{W}$. Through equation 64,

$$\sum_j \|A_j\|^2 \le \eta^2 \Big(4n\big(\|\nabla f(\bar{\mathbf{x}}^t)\|^2 + \delta^2\big) + 2L^2\Delta_t\Big). \tag{73}$$

For $B_j$, let $\Delta\mathbf{G}^t := \mathbf{G}^{t-1} - \mathbf{G}^t$ (row $k$ is $\mathbf{g}_k^{t-1} - \mathbf{g}_k^t$). Then

$$\sum_j \|B_j\|^2 = \eta^2 \|\mathbf{W}\Delta\mathbf{G}^t\|_F^2 \le \eta^2 \sum_{k=1}^n \|\mathbf{g}_k^{t-1} - \mathbf{g}_k^t\|^2 \le \eta^2 L^2 \sum_{k=1}^n \|\mathbf{d}_k^t\|^2 \tag{74}$$

Now we bound $C_j$:

$$\sum_{j=1}^{n} \|\mathbf{C}_j\|^2 = \|(\mathbf{I}-\mathbf{W})\mathbf{X}^{t-1}\|_F^2 = \|(\mathbf{W}-\mathbf{Q})(\mathbf{I}-\mathbf{Q})\mathbf{X}^{t-1}\|_F^2 \leq \|\mathbf{W}-\mathbf{Q}\|_2^2 \|(\mathbf{I}-\mathbf{Q})\mathbf{X}^{t-1}\|_F^2 = \lambda\,\Delta_{t-1}. \quad (75)$$

For $D_j$, $L$-smoothness and $\|\xi_k^{t-1}\| \leq \rho$, we have:

$$\|\widetilde{\mathbf{g}}_k^{t-1} - \mathbf{g}_k^{t-1}\| \leq L\rho \quad (76)$$

Combine equation 73-76:

$$\sum_j \|\mathbf{d}_j^t\|^2 \leq 4\eta^2\Big(4n\big(\|\nabla f(\bar{\mathbf{x}}^t)\|^2+\delta^2\big)+2L^2\Delta_t\Big) + 4\eta^2 L^2 \sum_j \|\mathbf{d}_j^t\|^2 + 4\lambda\Delta_{t-1} + 4L^2\rho^2$$
$$\big(1-4\eta^2 L^2\big)\sum_j \|\mathbf{d}_j^t\|^2 \leq 16\eta^2 n\big(\|\nabla f(\bar{\mathbf{x}}^t)\|^2+\delta^2\big) + 8\eta^2 L^2\Delta_t + 4\lambda\Delta_{t-1} + 4L^2\rho^2 \quad (77)$$

Assuming a standard stepsize condition $4\eta^2 L^2 \leq \frac{1}{2}$ (so $(1-4\eta^2 L^2)^{-1} \leq 2$) we get

$$\sum_j \|\mathbf{d}_j^t\|^2 \leq 32\,\eta^2 n\big(\|\nabla f(\bar{\mathbf{x}}^t)\|^2+\delta^2\big) + 16\,\eta^2 L^2\Delta_t + 8\lambda\,\Delta_{t-1} + 8L^2\rho^2 \quad (78)$$

Substituting the above result in equation 58:

$$\|A\|^2 \leq \Big(32\,\eta^4 n L^2\big(\|\nabla f(\bar{\mathbf{x}}^t)\|^2+\delta^2\big) + 16\,\eta^4 L^4\Delta_t + 8\lambda\eta^2 L^2\,\Delta_{t-1} + 8\eta^2 L^4\rho^2\Big) \quad (79)$$

Substituting equation 79, 65, 68 and 69 in equation 57:

$$\|\mathbf{d}_i^t - \eta\bar{\mathbf{g}}^t\|^2 \leq 4\Big(32\,\eta^4 n L^2\big(\|\nabla f(\bar{\mathbf{x}}^t)\|^2+\delta^2\big) + 16\,\eta^4 L^4\Delta_t + 8\lambda\eta^2 L^2\,\Delta_{t-1} + 8\eta^2 L^4\rho^2\Big)$$
$$+ 4\eta^2\lambda\Big[4n\big(\|\nabla f(\bar{\mathbf{x}}^t)\|^2+\delta^2\big) + 2L^2\,\Delta_t\Big] + 4\lambda\,\Delta_{t-1} + 4\eta^2 L^2\rho^2 \quad (80)$$

Collecting coefficients yields

$$\|\mathbf{d}_i^t - \eta\bar{\mathbf{g}}^t\|^2 \leq \underbrace{\Big(128\,\eta^4 L^2\,n + 16\,\eta^2\lambda\,n\Big)}_{C_g}\ \Big(\|\nabla f(\bar{\mathbf{x}}^t)\|^2 + \delta^2\Big)$$
$$+ \underbrace{\Big(64\eta^4 L^4 + 8\lambda\eta^2 L^2\Big)}_{C_\Delta}\Delta_t + \underbrace{(32\eta^2 L^2 + 4)}_{C_\Delta'}\lambda\Delta_{t-1} + \underbrace{(32L^2+4)L^2\eta^2\rho^2}_{C_\rho} \quad (81)$$

$$\|\mathbf{d}_i^t - \eta\bar{\mathbf{g}}^t\|^2 \leq C_g\Big(\|\nabla f(\bar{\mathbf{x}}^t)\|^2 + \delta^2\Big) + C_\Delta\,\Delta_t +; C_\Delta'\Delta_{t-1} + C_\rho, \quad (82)$$

with the explicit constants $C_g, C_\Delta, C_\Delta', C_\rho$ as above.

Averaging over total clients $n$ and training iterations $T$:

$$\frac{1}{T}\sum_{t=1}^{T}\frac{1}{n}\sum_{i=1}^{n}\|\mathbf{d}_i^t - \eta\bar{\mathbf{g}}^t\|^2 \leq C_g\Big(\frac{1}{T}\sum_{t=1}^{T}\|\nabla f(\bar{\mathbf{x}}^t)\|^2+\delta^2\Big) + C_\Delta\frac{1}{T}\sum_{t=1}^{T}\Delta_t + C_\Delta'\frac{1}{T}\sum_{t=1}^{T}\Delta_{t-1} + C_\rho, \quad (83)$$

Now similar to main paper, let $\eta = \mathcal{O}(\sqrt{n/T})$ and $\rho = \mathcal{O}(1/\sqrt{T})$.

From the result of Lemma 3:

$$\frac{1}{T}\sum_{t=1}^{T}\Delta_t = \mathcal{O}\Big(\frac{1}{T}\Big) \tag{84}$$

From Corollary 2:

$$\frac{1}{T}\sum_{t=1}^{T}\|\nabla f(\bar{\mathbf{x}}^t)\|^2 = \mathcal{O}\Big(\frac{1}{nT}\Big) \tag{85}$$

$$
\begin{aligned}
C_g &= \mathcal{O}\Big(\frac{1}{T^2} + \frac{1}{T}\Big) = \mathcal{O}\Big(\frac{1}{T}\Big)\\
C_\Delta &= \mathcal{O}\Big(\frac{1}{T^2} + \frac{1}{T}\Big) = \mathcal{O}\Big(\frac{1}{T}\Big)\\
C'_\Delta &= \mathcal{O}\Big(\frac{1}{T} + 1\Big) = \mathcal{O}(1)\\
C_\rho &= \mathcal{O}\Big(\frac{1}{T^2}\Big)
\end{aligned}
\tag{86}
$$

Putting together the above equations, for a fixed $n << T$:

$$
\begin{aligned}
\frac{1}{T}\sum_{t=1}^{T}\frac{1}{n}\sum_{i=1}^{n}\|\mathbf{d}_i^t - \eta\bar{\mathbf{g}}^t\|^2 &\leq \mathcal{O}\Big(\tfrac{1}{T}\Big)\Big(\tfrac{1}{nT} + \delta^2\Big) + \mathcal{O}\Big(\tfrac{1}{T}\Big)\tfrac{1}{T} + \mathcal{O}\Big(\tfrac{1}{T}\Big) + \mathcal{O}\Big(\tfrac{1}{T^2}\Big)\\
&= \mathcal{O}\Big(\tfrac{1}{T}\Big) + \mathcal{O}\Big(\tfrac{1}{nT^2}\Big) + \mathcal{O}\Big(\tfrac{1}{T}\Big) + \mathcal{O}\Big(\tfrac{1}{T^2}\Big)\\
&= \mathcal{O}\Big(\tfrac{1}{T}\Big)
\end{aligned}
\tag{87}
$$

**Interpretation.** Perturbation deviation quantifies how closely each client's displacement $\mathbf{d}_i^t$ aligns with the global gradient $\eta\bar{\mathbf{g}}^t$, and is controlled by the following key factors:

$$\|\mathbf{d}_i^t - \eta\bar{\mathbf{g}}^t\|^2 \leq C_g\big(\|\nabla f(\bar{\mathbf{x}}^t)\|^2 + \delta^2\big) + C_\Delta\,\Delta_t + C'_\Delta\,\Delta_{t-1} + C_\rho,$$

The bound shows that this deviation is governed by the global gradient magnitude, the consensus error among clients, and the SAM perturbation radius. As training progresses, both the global gradient norm and the consensus error $\Delta_t$ decay, while the perturbation effect remains bounded and scales with the decaying learning rate. Consequently, the deviation diminishes at rate $\mathcal{O}(1/T)$, indicating that each client's local displacement asymptotically tracks the global descent trend, validating our approximated global direction as a faithful proxy without requiring any global synchronization.

**Implementation note.** In our implementation (Algorithm 1), each client forms the perturbation direction as a convex combination of its local and approximated global directions:

$$\xi_i^t = \rho\big(s.\frac{\mathbf{d}_i^t}{\|\mathbf{d}_i^t\|} + (1-s).\frac{\mathbf{g}_i^t}{\|\mathbf{g}_i^t\|}\big), \tag{88}$$

where $\rho$ is the perturbation radius and $s$ controls the emphasis on the global vs local direction.

Since $\mathbf{d}_i^t = \mathbf{x}_i^{t-1} - \mathbf{x}_i^t$ already represents the model update that includes the step size $\eta$, we include $\eta$ only with the global gradient term in Lemma 5. In other words, $\eta$ is implicit in $\mathbf{d}_i^t$ but explicit in $\eta\bar{\mathbf{g}}^t$, ensuring that both quantities are comparable in scale.

### E.2 Empirical Validation

To validate the theoretical result of Lemma 5, we compare our practical implementation of *GFlat*, which uses the locally approximated global direction $\mathbf{d}_i^t$, against its oracle counterpart *GFlat (Oracle)* in Table 8,

which directly uses the true global gradient $\bar{\mathbf{g}}^t$ for the SAM ascent step. As expected from the bound on perturbation deviation, the oracle variant achieves consistently higher accuracy across all settings, serving as an upper bound on the achievable performance under perfect global coordination. However, the small performance gap between the two confirms that our locally approximated direction effectively tracks the global gradient without requiring global synchronization.

Table 8: Test accuracy (%) on CIFAR-10 with ResNet-20 under different levels of heterogeneity ($\alpha$). GFlat (Oracle) denotes the variant that uses the true global gradient $\bar{\mathbf{g}}^t$ in the SAM ascent step and serves as an upper bound on achievable performance, while GFlat uses the locally approximated global direction.

| Clients | Method | CIFAR-10 (ResNet20) | |
| --- | --- | --- | --- |
| | | $\alpha = 0.01$ | $\alpha = 0.001$ |
| 10 | *GFlat* | $70.12 \pm 0.68$ | $62.29 \pm 0.66$ |
| | *GFlat (Oracle)* | $73.84 \pm 0.46$ | $66.41 \pm 0.34$ |
| 20 | *GFlat* | $56.21 \pm 1.33$ | $51.76 \pm 0.68$ |
| | *GFlat (Oracle)* | $57.41 \pm 0.45$ | $52.09 \pm 0.94$ |

### E.3 Alignment Analysis

We further quantify how well the update directions in *GFlat* align with the global gradient by measuring the deviation $\| s\,\mathbf{d}_i^t + (1-s)\mathbf{g}_i^t - \bar{\mathbf{g}}^t \|$ averaged over clients. For comparison, we compute the same metric for *SADDLe*, which relies solely on local gradients, i.e., $\|\mathbf{g}_i^t - \bar{\mathbf{g}}^t\|$. As shown in Figure 9, *GFlat* exhibits substantially smaller deviation, demonstrating that incorporating the approximated global direction improves curvature alignment across clients and leads to more coordinated updates, along with globally flat model as demonstrated in Figure 2.

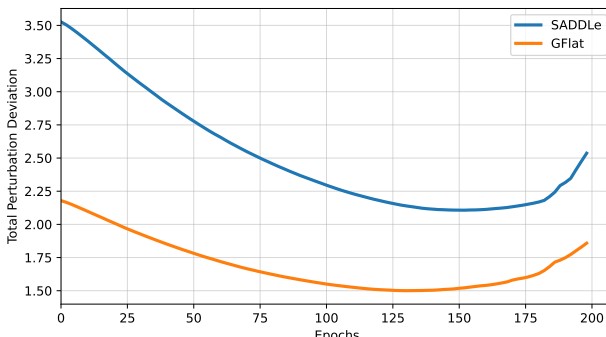

Figure 9: Total perturbation deviation between each client's update direction and the true global gradient $\bar{\mathbf{g}}^t$ on CIFAR-10 distributed across 10 clients with $\alpha = 0.01$. The smaller deviation observed for *GFlat* indicates stronger alignment with the global gradient, validating the theoretical perturbation deviation bound.

