# OpenReview forum: "Achieving Global Flatness in Decentralized Learning with Heterogeneous Data"
_TMLR — Accepted by TMLR_

### Review · Reviewer_SmoR · 2025-10-16

**Summary Of Contributions:**

The paper introduces GFlat, a decentralized optimization method intended to achieve "global flatness" of the loss landscape when data across clients are heterogeneous (non-IID). The main idea is to modify the Sharpness-Aware Minimization (SAM) optimizer at each client by adding an approximation of the global update direction, estimated from model differences between consecutive iterations. This modification requires no additional communication and is claimed to improve generalization. The authors also provide a convergence analysis and experimental results on several datasets and network topologies.

**Additional Comments:**

- The paper’s structure and motivation are acceptable, but the writing and technical precision can be improved.
- The connection between the proposed perturbation scheme and actual “global flatness” is not substantiated theoretically.

**Audience:**

Yes

**Audience Explanation:**

The topic: decentralized optimization with non-IID data, is relevant to TMLR’s audience.
The experimental section seems to demonstrate that the authors manage to address that issue.
However, the proposed modification is minor compared to existing SAM variants, and the theoretical content largely reuses standard proofs from decentralized SGD literature.

**Claims And Evidence:**

Yes

**Claims Explanation:**

The theoretical section proves a standard convergence rate that does not justify the claimed "global flatness" property.
Moreover, several notational and conceptual inconsistencies make parts of the analysis difficult to follow.
The exposition often lacks rigor, and key ideas such as how the approximation $d_t = x_{t-1} - x_t$ captures any meaningful global information remain speculative.

That being said, the experiments show accuracy improvements over SADDLe and other baselines, which seem to indicate that the proposed algorithm has the advertised effects.

**Requested Changes:**

**Critical:**

- Clarify notation and correct typographical errors.
   - In Eq. (3), I think that it should be $f_\rho(x) = \max_{\|\xi\|\le\rho} f(x+\xi)$.
   - Eq. (4): I think that it should read $\nabla f_\rho(x) \approx \nabla f (x + \rho g/ \| g\|) \dots$
   - Eq. (5): missing parentheses to make the max scope clear.
   - Eq. (6): use different symbols for $d$ (data) and $d$ (direction). It is currently confusing.

- Discuss why gradient momentum is used only in the first step where it is used to estimate the worst case point locally, and not also when you do the actual update.

- Theorem 1: explain whether $\epsilon$ is random or deterministic and whether a $\max_T$ is missing in its definition.

- Motivation and interpretation: provide a clearer argument for why the local difference $x_t - x_{t-1}$ conveys true global information; as it stands, this is not convincing. Indeed, it aggregates local information about the neighbors, but beyond that, I do not see why it translates to global information.

- Can you clarify a bit more the link between local flatness and global flatness? As for me, it might even be worse than simply saying "local flatness does not imply global flatness": aggregating local critical points might not even lead to a global critical point.

**Recommended (non-blocking):**



- Note that Assumption 2 trivially holds for $\sigma^2 = 2$ after normalization.

- Include more details on how non-IID partitions are created in the main text.

- Clarify the DPSGD acronym, which is usually understood as *Differentially Private* SGD, not *Decentralized Parallel* SGD.


- Explain the meaning of "orthogonal" in "orthogonal and compatible". If it simply means "independent of communication topology", say so explicitly.

---

> ### Author Response · Authors · 2025-11-09
> **Summary of Revisions and Additional Clarifications**
>
> We sincerely thank the reviewer for their detailed feedback and clarification requests. We have uploaded a revised manuscript incorporating all suggestions, with the corresponding changes highlighted in **blue**.
>
> We would like to highlight that our convergence analysis provides new theoretical insights specific to our approach of *locally approximating the global perturbation direction*. In particular, the approximation error in estimating the global perturbation appears as a higher-order term, whose influence on convergence diminishes as training progresses.
>
> Below, we briefly summarize the key changes made in response to each comment:
>
> 1. All typographical inconsistencies have been addressed in Equations 3-5. For denoting the minibatch data, we replaced $d$ with $\mathcal{B}$ to avoid confusion with model differences $\mathbf{d}$.
>
> 2. We presented results for Q-GFlat in Tables 2, 3, and 4 of the main paper, which integrates a momentum term into the gradient descent step. These results show that our proposed globally informed SAM ascent step is complementary to modifications in the descent step and consistently improves performance across all settings.
>
> 3. $\epsilon$ is the maximum error over training iterations as well as total clients. We have fixed the notation in Theorem 1.
>
> 4. We thank the reviewer for this insightful comment. Note that we present the eigen spectra of the Hessian in Figures 2 and 3 to highlight how our proposed approach results in a globally flat model, unlike existing techniques. To address the concern regarding why the locally computed direction in GFlat conveys global information and how it relates to global flatness, we have **added a dedicated Appendix E titled “Discussion on Local and Global Flatness”**. In this section, we first clarify that under IID data, the local and global Hessian spectra are nearly identical, but under heterogeneous (non-IID) data, the local curvature can deviate significantly from the global one. Optimizing for local flatness alone does not ensure global flatness, since locally “flat” regions may correspond to globally sharp directions. We attribute this misalignment to the fact that each client estimates its worst-case perturbation using only its own local gradient, which captures limited curvature information. Ideally, one would instead compute perturbations based on the global gradient ($\bar{\mathbf{g}}^t=1/n\sum_{i=1}^n \mathbf{g}_i^t$), ensuring that every client’s update direction is consistent with the global curvature. While this would align local SAM steps with global flatness, such synchronization
> fundamentally violates the decentralized setup, requiring full communication among all clients at every iteration.
> To overcome this, we introduce a locally approximated global direction $\mathbf{d}_i^t$, which allows each client to estimate $\bar{\mathbf{g}}^t$ using model differences across consecutive training iterations. This approximation enables clients to incorporate a proxy of the global update into their local SAM objectives, effectively aligning local and global flatness without incurring any communication overhead. We refer to the discrepancy between the approximated and true global directions, $||\mathbf{d}_i^{t}-\eta \bar{\mathbf{g}}^{t}||$, as the perturbation deviation, which quantifies how closely each client's local displacement tracks the global gradient direction. Lemma 5 in the revised paper formalizes this intuition by showing that the perturbation deviation decays asymptotically at a rate of $\mathcal{O}(1/T)$. Empirically, we compare our practical implementation of GFlat, which uses the locally approximated direction $\mathbf{d}_i^{t}$, with an oracle variant GFlat(Oracle) that directly uses the true global gradient. As shown in Table 8, the oracle achieves slightly higher accuracy, confirming it acts as an upper bound, while the small performance gap demonstrates that our approximation effectively tracks the global gradient without synchronization. Additionally, we measure the alignment between each client’s effective update direction and the true global gradient, quantified by $||s\mathbf{d}_i^{t}+(1-s)\mathbf{g}_i^{t}-\bar{\mathbf{g}}^{t}||$, and compare it against SADDLe, which relies solely on local gradients ($||\mathbf{g}_i^{t}-\bar{\mathbf{g}}^{t}||$). As shown in Figure 9, GFlat exhibits consistently smaller deviation, indicating improved curvature alignment across clients and a globally flatter model, as also reflected in Figure 2.
>
> 5. We have added more details regarding our non-IID data partitions in Section 6.1.
>
> 6. We have clarified the DPSGD acronym in both the introduction and background sections to avoid confusing it with Differentially Private SGD.
>
> 7. By the phrase "orthogonal and compatible" we wanted to highlight that GFlat is complementary to existing decentralized techniques for non-IID data and is agnostic to chosen graph topology. We have revised this sentence in the paper for further clarity.

---

### Review · Reviewer_gHAo · 2025-10-21

**Summary Of Contributions:**

The paper builds on sharpness aware minimization to obtain an optimization algorithm for decentralized learning. Sharpness aware optimization has minimax objective function which aims to obtain a local minimum with largest flat neighborhood, in the sense that function varies slowest in the neighborhood of local minimum. In decentralized learning scenarios, in order to obtain the flat neighborhood around a point, one need to obtain gradients of global objective function. There is a number of factors like privacy, computational and communication overheads which make the computation of global gradients impossible. Therefore, the paper combines the ideas from previous work [1] to estimate and update perturbation using local gradients and model differences.

[1] Lin T, Karimireddy SP, Stich S, Jaggi M. Quasi-global Momentum: Accelerating Decentralized Deep Learning on Heterogeneous Data. InInternational Conference on Machine Learning 2021 Jul 1 (pp. 6654-6665). PMLR.

Strengths:
The paper is very well written and clearly motivates the problem.

Weaknesses:
The paper has minimal novelty.

**Audience:**

Yes

**Audience Explanation:**

The paper deals with problem of decentralized learning, which is an important practical problem in machine learning.

**Broader Impact Concerns:**

No.

**Claims And Evidence:**

Yes

**Claims Explanation:**

Paper is very well written and well-motivated, and the claims made in the seem to be accurate.

**Requested Changes:**

The paper is well organized and clearly written. I have only a few minor suggestions.
In Equation (2), the notations $d_i^t$ and $\boldsymbol{d}_i^t$ seem to represent two distinct quantities; please consider changing one of them to avoid confusion.
In addition, it would be helpful to elaborate on the update rule in Step 8 of Algorithm 1 and clarify how it differs from the traditional update rule in Equation (2) for DPSGD.

---

> ### Author Response · Authors · 2025-11-09
> **Response to Reviewer gHAo**
>
> We sincerely thank the reviewer for their time and thoughtful feedback. We clarify that our work makes a novel contribution to decentralized learning under non-IID data partitions by *explicitly characterizing and addressing the discrepancy between local and global flatness*, a phenomenon that we both theoretically analyze and empirically verify.  This perspective provides new insight into why optimizing solely for local flatness, as done in prior approaches, can lead to poorly generalizing models under heterogeneous data distributions.
>
> Our proposed method, GFlat, introduces a *locally approximated global perturbation direction* that achieves stronger consensus and global flatness *without any additional communication or computational overhead*. We view this as a key strength: it combines conceptual simplicity with consistent empirical gains (Tables 1–5).  Overall, the contribution lies in offering a new theoretical and empirical understanding of flatness discrepancy in decentralized learning and demonstrating that this insight translates into measurable performance improvements.
>
> Below, we summarize the key changes made in response to each comment:
>
> **Notations:** We have changed the data notation $d$ to $\mathcal{B}$ to avoid confusion and revised the paper accordingly.
>
> **Elaborating the update rule in Algorithm 1:** The key difference between traditional DPSGD and our proposed GFlat lies in the local optimization step performed before model averaging. In DPSGD, each client updates its parameters directly using the local gradient $\mathbf{g}_i^t$, followed by a weighted averaging with its neighbors as shown in Equation 2. In contrast, GFlat introduces an additional perturbation-aware ascent step, where each client first perturbs its parameters along a locally approximated global direction before performing the descent update. This yields an intermediate model $\mathbf{x}_i^{t+1/2}$, which is then mixed with those of neighboring clients as shown in line 8 in Algorithm 1.
> Consequently, GFlat differs from DPSGD only in the local optimization step, while the communication and aggregation mechanisms remain identical.

---

### Review · Reviewer_ZN39 · 2025-10-29

**Summary Of Contributions:**

This paper presents GFlat -- a new decentralized learning algorithm that improves generalization in non-IID data settings through seeking global flatness. Authors propose a simple strategy that improves upon SADDLe -- a method that only achieves local flatness. Both methods define the objective as a sharpness-aware minimization problem for decentralized setting. However, GFlat redefines the perturbation $\xi$ for each client $i$ as a combination of local and global perturbations. To quantify the flatness authors utilize a proxy for loss curvature -- ratio of largest to the 5th largest eigenvalue $(\lambda_{max}/\lambda_5)$. Authors also provide a theoretical analysis of GFlat that shows convergence rate similar to existing literature. Assumptions stated for the analysis are reasonable and standard throughout similar methods.

Authors conducted experiments on image classification task with 3 DNN models and 2 client topologies, and compare GFlat to SADDLe and DPSGD. Results show consistent improvements over other methods.

Additionally, paper has a coherent structure and a comprehensive introduction to the problem and related literature.

---

Strengths:
1. Problem is clearly defined with a relevant motivation.
2. Presented methodology has a clear advantage over existing methods which is shown through numerical experiments.
3. GFlat incurs no additional communication overhead and only requires $\mathcal{O}(m)$ memory at each client.
4. Scaling factor $s$ in the redefined perturbation $\xi$ allows shifting emphasis on global or local perturbation, which makes the method flexible to different degrees of data heterogeneity.

Weaknesses:
1. Code used for experiments is not included with the submission which makes it hard to reproduce the numerical results.

**Audience:**

Yes

**Audience Explanation:**

I found the topic of this paper to be relevant for TMLR's audience based on other submissions accepted to TMLR that were studying similar research directions.

**Broader Impact Concerns:**

I have not found any ethical concerns in this work.

**Claims And Evidence:**

Yes

**Claims Explanation:**

I have found authors' claims to be accurate based on presented numerical experiments.

**Requested Changes:**

Include some type of reference to the code repository that was used to conduct the numerical experiment.

---

> ### Author Response · Authors · 2025-11-09
> **Response to Reviewer ZN39**
>
> We sincerely thank the reviewer for their time and detailed feedback.
>
> We would like to clarify that the complete implementation for all experiments is provided as a zipped folder under the "Supplementary Material". The package includes a `README.md` file with detailed instructions for environment setup and bash scripts to reproduce all experiments.  Hence, the *code is fully available and reproducible as part of the submitted supplementary files*. We will also publicly release the code upon acceptance to facilitate further research by the decentralized community.

---

### Decision · Action_Editor_AfRG · 2025-12-12

**Recommendation:** Accept as is

**Audience:**

Yes

**Audience Explanation:**

The paper proposes a new decentralized learning algorithm. A noticeable part of TMLR's audience will be interested in the findings of this paper.

**Claims And Evidence:**

Yes

**Claims Explanation:**

The claims are supported by rigorous proofs and numerical experiments. All reviewers agree on this.